# The promise and peril of comparing fluorescence lifetime in biology revealed by simulations

**Pingchuan Ma[1,2], Peter Chen[1,3], Scott Sternson[4], Yao Chen[1]\***

[1]Department of Neuroscience, Washington University in St. Louis, St. Louis, United States; [2]Ph.D. Program in Neuroscience, Washington University in St. Louis, St. Louis, United States; [3]Master's Program in Biomedical Engineering, Washington University in St. Louis, St. Louis, United States; [4]Department of Neuroscience, Howard Hughes Medical Institute, University of California, San Diego, San Diego, United States

## eLife Assessment

This study presents an **important** computational framework, FLiSimBA (Fluorescence Lifetime Simulation for Biological Applications), for modeling experimental limitations in Fluorescence Lifetime Imaging Microscopy (FLIM). FLiSimBA is readily available in MATLAB and Python, enables users to simulate effects of noise and varying sensor expression levels, and provides practical guidance for both lifetime imaging experiments and biosensor development. The analyses are robust, and the evidence supporting the tool's utility in distinguishing between multiple lifetime signals is **compelling**, indicating strong potential for multiplexed dynamic imaging. However, users should also consider that the tool's effectiveness depends on the suitability of a two-component discrete exponential model.

**\*For correspondence:**
Yaochen@wustl.edu

**Competing interest:** The authors declare that no competing interests exist.

**Abstract** Signaling dynamics are crucial in biological systems, and biosensor-based real-time imaging has revolutionized their analysis. Fluorescence lifetime imaging microscopy (FLIM) excels over the widely used fluorescence intensity imaging by allowing the measurement of absolute signal levels independent of sensor concentration. This capability enables the comparison of signaling dynamics across different animals, body regions, and timeframes. However, FLIM's advantage can be compromised by factors like autofluorescence in biological experiments. To address this, we introduce FLiSimBA, a flexible computational framework for realistic F̲luorescence L̲ifetime S̲imulation for B̲iological A̲pplications. Through simulations, we analyze the signal-to-noise ratios of fluorescence lifetime data, determining measurement uncertainty and providing necessary error bars for lifetime measurements. Furthermore, we challenge the belief that fluorescence lifetime is unaffected by sensor expression and establish quantitative limits to this insensitivity in biological applications. Additionally, we propose innovations, notably multiplexed dynamic imaging that combines fluorescence intensity and lifetime measurements. This innovation can transform the number of signals that can be simultaneously monitored, thereby enabling a systems approach in studying signaling dynamics. Thus, by incorporating different factors into our simulation framework, we uncover surprises, identify limitations, and propose advancements for fluorescence lifetime imaging in biology. This quantitative framework supports rigorous experimental design, facilitates accurate data interpretation, and paves the way for technological advancements in fluorescence lifetime imaging.

## Introduction

FLIM and fluorescence lifetime photometry (FLiP) are powerful methods for revealing the dynamics of biological signals (*Bastiaens and Squire, 1999*; *Becker, 2012*; *Brinks et al., 2015*; *Chen et al., 2017*; *Chen et al., 2014*; *Lakowicz et al., 1992*; *Laviv et al., 2020*; *Lazzari-Dean et al., 2019*; *Lee et al., 2019*; *Lee et al., 2021*; *Ma et al., 2018*; *Ma et al., 2024*; *Massengill et al., 2022*; *Mongeon et al., 2016*; *Tang and Yasuda, 2017*; *van der Linden et al., 2021*; *Yasuda, 2006a*; *Yasuda et al., 2006b*; *Zhang et al., 2021b*; *Zheng et al., 2015*). Fluorescence lifetime refers to the time between excitation of a fluorophore and emission of light. Compared with intensity-based imaging, the greatest advantage of fluorescence lifetime is its insensitivity to fluorophore concentration. As a result, fluorescence lifetime can be used to compare the dynamics of biological signals across animals or over long periods of time despite changes in sensor expression levels (*Laviv et al., 2020*; *Ma et al., 2024*). Furthermore, it offers the potential to quantitate absolute values of biological signals because of its quantitative nature and its insensitivity to sensor expression. Because of these advantages, FLIM-compatible sensors are being actively developed and FLIM has been increasingly adopted to elucidate the dynamics of many types of biological signals over multiple time scales.

Although fluorescence lifetime is independent of sensor expression when only the sensor is present, in biology experiments, this very advantage of FLIM breaks down when multiple other factors are present. These include autofluorescence (natural emission of light by biological tissue after light absorption), background light (e.g. ambient light), dark current, and afterpulse of the photomultiplier tube (PMT) (*Akgun et al., 2008*; *Georgakoudi and Quinn, 2023*; *Malak et al., 2022*). As sensor expression varies, the relative contribution of sensor fluorescence over these other sources of light or electrical noise varies correspondingly, leading to an apparent change in fluorescence lifetime. Thus, to harness the power and correctly interpret the results of FLIM and FLiP experiments in biological tissue, it is critical to quantitatively understand the regime in which fluorescence lifetime varies with sensor expression, and the range in which sensor expression does not significantly alter lifetime measurements. Furthermore, these additional factors introduce bias and noise to fluorescence lifetime measurements. As innovation pushes the technological boundary to image larger fields of view at higher speeds (*Bowman et al., 2023*; *Lodder et al., 2025*; *Raspe et al., 2016*; *Shcheslavskiy et al., 2018*; *Zhang et al., 2021a*), it is critical to understand how these factors contribute to signal-to-noise (SNR) ratio, and how many photons are required to achieve a certain SNR in biological settings.

An effective tool for exploring how experimental parameters contribute to outcomes is simulation. Both analytical and simulation methods have provided insights into issues such as the SNR (*Esposito et al., 2007*; *Esposito et al., 2008*; *Gerritsen et al., 2002*; *Köllner and Wolfrum, 1992*; *Kumar, 2012*; *Lakowicz, 2006*; *Nasser and Meller, 2022*; *Netaev et al., 2022*; *Philip and Carlsson, 2003*; *Roethlein et al., 2015*; *Trinh and Esposito, 2021*; *Turton et al., 2003*; *Walsh et al., 2016*; *Xiao et al., 2021*; *Yasuda, 2006a*). However, prior work usually assumes the presence of sensor fluorescence only, without considering the important contributions to noise and bias due to other factors, such as autofluorescence and afterpulse of the PMT. Consequently, these simulations are useful in vitro but are not readily applicable in biological settings. Therefore, to understand how experimental conditions influence lifetime estimates for biological applications, simulations with realistic and, ideally, measured data must be performed.

Here, we introduce <u>Fl</u>uorescence <u>Li</u>fetime <u>Sim</u>ulation for <u>B</u>iological <u>A</u>pplications (FLiSimBA) and use it to quantitatively define the potential and limitations of fluorescence lifetime experiments in biological settings. FLiSimBA is a flexible platform designed for realistic simulation of fluorescence lifetime data with empirically determined parameters through time-correlated single photon counting (TCSPC). FLiSimBA recapitulates experimental data. Using FLiSimBA, we determined the photon requirements for minimum detectable differences in fluorescence lifetime, thus providing realistic estimates of the SNR in biological tissue and the necessary quantification of measurement uncertainty for correct data interpretation. Furthermore, we challenge the conventional view that fluorescence lifetime is insensitive to sensor expression levels and establish quantitative limits of insensitivity, defining the potential and limits of comparing fluorescence lifetime measurements in biology. Finally, we use FLiSimBA to propel innovation in FLIM by assessing the impact of hardware improvement on the SNR, quantifying the value of developing sensors with spectra unaffected by autofluorescence, and specifying sensor characteristics that would greatly expand the power of simultaneous real-time measurements of multiple signals (multiplexing) with both intensity and lifetime properties of sensors.

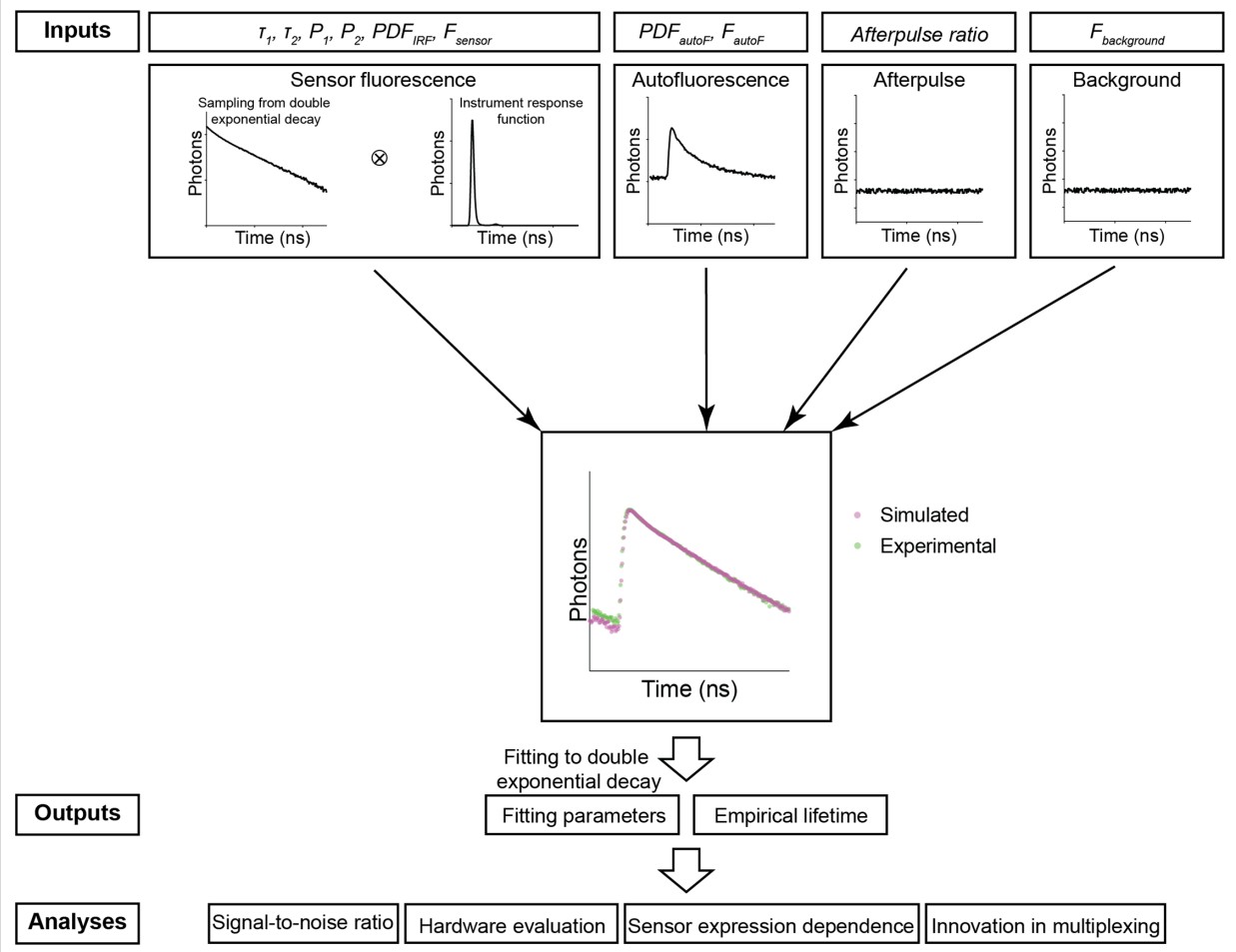

**Figure 1.** The simulation procedure of Fluorescence Lifetime Simulation for Biological Applications (FLiSimBA). The sensor fluorescence lifetime distribution of a FRET-based sensor was modeled as a double exponential decay, as shown in *Equation 1* ($\tau_1$=2.14 ns, $\tau_2$=0.69 ns) and the lifetimes of photons were sampled with replacement. After the sensor fluorescence was convolved with the probability density function (PDF) of the instrument response function (IRF), the following were added to produce the final simulated data: autofluorescence (autoF) empirically measured from brain tissue with photon number $F_{autoF}$, afterpulse of the photomultiplier tube (PMT), and background signal (consisting of the dark current of the PMT and light leak) with photon number $F_{background}$. The histograms of the final simulated data were similar to those of the experimental data. The simulation was repeated 500 times under each $P_1$ and sensor fluorescence photon number ($F_{sensor}$) condition. The fitted $P_1$ (based on a double exponential decay fitting of the final simulated data) and empirical lifetime were used for subsequent analyses and evaluations.

Our results show that considering relevant contributors and noise to fluorescence lifetime measurements in biological contexts does not merely add detail. Rather, these quantitative changes produce qualitatively different conclusions that require careful consideration to avoid misinterpretation of data and to harness the potential of FLIM. Thus, we provide a versatile tool for simulating experimental conditions, a quantitative framework for evaluating fluorescence lifetime results, and insights into the potential and limitations of fluorescence lifetime measurements in biological applications.

## Results

### Simulation of fluorescence lifetime data in biological tissue

To realistically mimic the fluorescence lifetime in biological tissue, we simulated contributions from sensor fluorescence, autofluorescence, afterpulse, and background, which is composed of both a small amount of light leakage and the dark current of the photon detectors (*Figure 1*).

For sensor fluorescence, we sampled a specific number of photons ($F_{sensor}$) with replacement from an ideal distribution of photon lifetimes. In the examples in this study, we used the lifetime distribution

of the FLIM-compatible A Kinase Activity Reporter (FLIM-AKAR), a Förster resonance energy transfer (FRET)-based biosensor that measures the activity of protein kinase A (PKA) in a variety of biological contexts, including brain slices and freely moving animals (*Chen et al., 2014*; *Chen et al., 2017*; *Lee et al., 2019*; *Lee et al., 2021*; *Tilden et al., 2024*). The fluorescence lifetime of FLIM-AKAR follows a double exponential decay defined by the following equation:

$$F\left(t\right) = F_0 * (P_1 * e^{\left(-\frac{t}{\tau_1}\right)} + P_2 * e^{\left(-\frac{t}{\tau_2}\right)}) \tag{1}$$

where $F(t)$ is the number of photons that arrive at time t, $F_0$ is the number of photons at time 0, $\tau_1$, and $\tau_2$ are time constants corresponding to lifetime distributions of the donor fluorophore that is either free or undergoing FRET, and $P_1$ and $P_2$ are the proportions of the donor fluorophores in these two states (*Figure 1*). Following sampling, we convolved the lifetime histogram with the probability density function (PDF) of an instrument response function (IRF) to account for instrument noise.

Subsequently, we added a specific number of sampled photons ($F_{autoF}$) from an autofluorescence curve, whose distribution was determined via fluorescence measurements in brain tissue without any sensor expression. The autofluorescence lifetime histogram exhibited faster decay than that of FLIM-AKAR (*Figure 2—figure supplement 1A*). After that, we added the afterpulse of the PMT, long-lasting signals from ionization of residual gas inside the PMT following a photon detection event (*Akgun et al., 2008*). The afterpulse was modeled as an even lifetime distribution with the amount of signal as a fraction of sensor fluorescence. Finally, we added a specific number of background signals ($F_{background}$) that was empirically determined from measurements, accounted for largely by ambient light leakage and the dark current of the PMT. We generated 500 simulated fluorescence lifetime histograms for each $P_1$ and sensor photon count (for additional details, see Materials and methods).

The simulated histogram closely matched the experimental histogram (*Figure 1*). Following histogram generation, we used two commonly used fluorescence lifetime metrics to evaluate the simulated data (*Chen et al., 2017*; *Harvey et al., 2008*; *Lee et al., 2021*; *Ma et al., 2024*; *Mongeon et al., 2016*; *Yasuda et al., 2006b*). First, fluorescence lifetime data were evaluated with empirical lifetimes, defined as the average lifetime of all photons:

$$empirical\,lifetime = \frac{\sum\left(F\left(t\right) * t\right)}{\sum F\left(t\right)} \tag{2}$$

in which $t$ is the lifetime of photons arriving at a specific time channel, and $F(t)$ is the photon count from that time channel. Additionally, simulated histograms were fitted with a double exponential decay equation with Gauss-Newton nonlinear least-square fitting algorithm:

$$F(t) = \left[F_0 * \left(P_1 * e^{-\frac{t}{\tau_1}} + P_2 * e^{-\frac{t}{\tau_2}}\right) + SHG\right] \otimes IRF + F_{background} \tag{3}$$

in which $F_0$ is photon count from sensor fluorescence at time 0, $F_{background}$ is background signal, SHG is second harmonic generation that is added at time 0, and ⊗ represents convolution. Both experimental and simulated data showed good fitting (*Figure 2—figure supplement 1B*). For subsequent analyses and evaluations, empirical lifetime and fitted $P_1$, corresponding to the proportion of slower decay (free donor fluorophore) were used (*Figure 1*).

## Bias and noise introduced by different sources of signals

To understand how different fluorescence and electronic sources contribute to bias and noise, we analyzed the fitted $P_1$ and empirical lifetime across a range of values for the simulated $P_1$ and sensor photon counts. Autofluorescence (autoF) decreased the fitted $P_1$ and empirical lifetime, consistent with the faster decay of autofluorescence compared with that of FLIM-AKAR sensor fluorescence (*Figure 2*, *Figure 2—figure supplement 1*; *Figure 2B*: p<0.0001, sensor +autoF vs sensor only under all sensor photon number conditions for both fitted $P_1$ and empirical lifetime). Further addition of afterpulse and background did not significantly change the fitted $P_1$ (*Figure 2*; *Figure 2B*: p>0.6, final simulated data vs sensor + autoF condition for all sensor photon number conditions), consistent with consideration of the background term during the fitting procedure. In contrast, afterpulse and

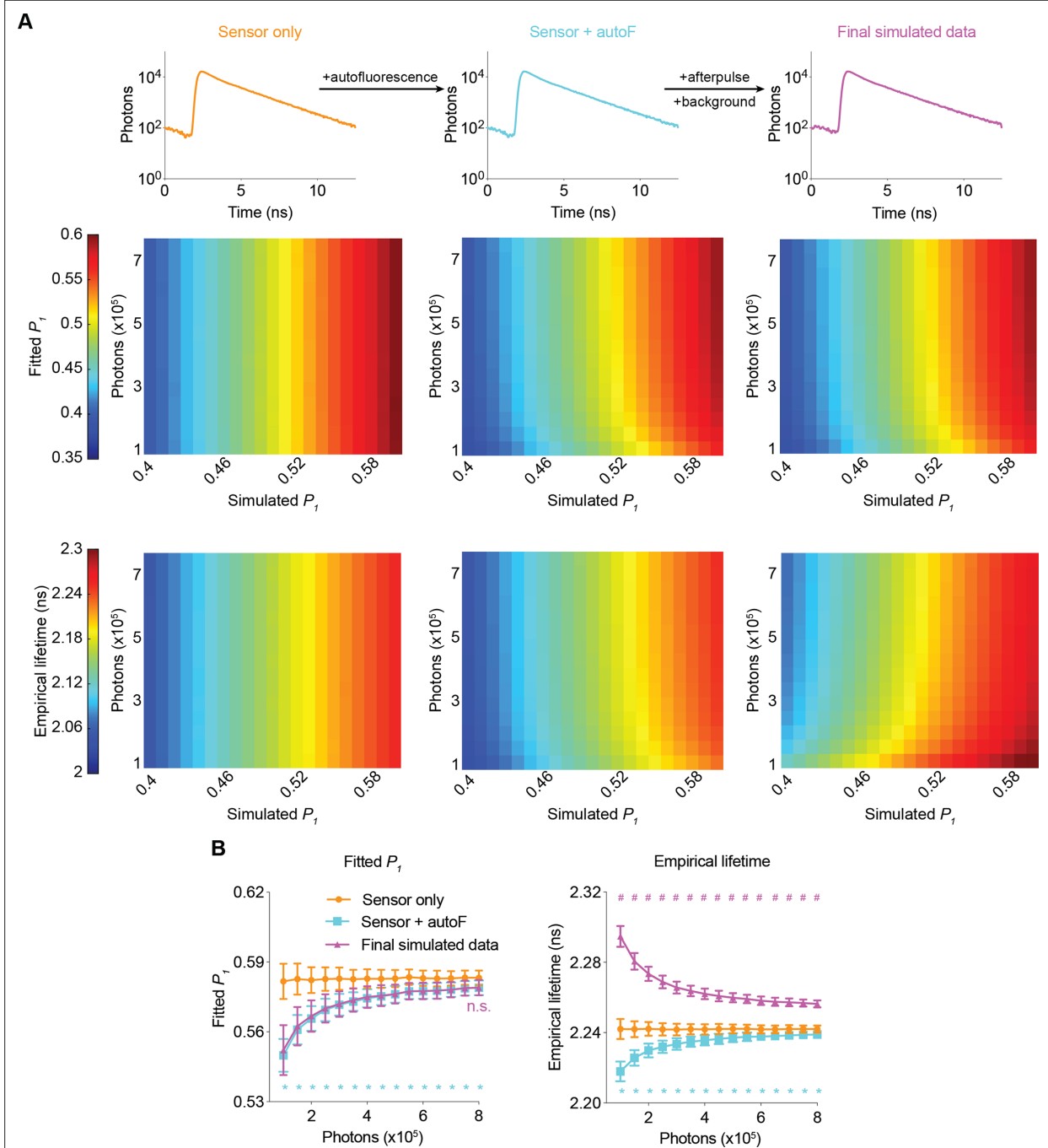

**Figure 2.** Simulated fluorescence lifetime in brain tissue. (**A**) Simulated example histograms (upper panels) and heatmaps (middle and lower panels) showing the average fitted $P_1$ and empirical lifetime of the simulated data across a range of $P_1$ and sensor photon number conditions. Fluorescence lifetime histograms were simulated with the sensor only, sensor + autofluorescence (autoF), and sensor + autofluorescence + afterpulse + background (final simulated data). (**B**) Summaries of fitted $P_1$ (left) and empirical lifetime (right) with simulated $P_1$=0.5854. *$p<0.05$ for sensor + autoF (cyan) vs sensor only (orange); n.s. not significant for final simulated data (purple) vs sensor + autoF (cyan); #$p<0.05$ for final simulated data (purple) vs sensor + autoF (cyan). Two-way ANOVA with Šídák's multiple comparisons test. n=500 simulations. The data are represented as means and standard deviations.

The online version of this article includes the following figure supplement(s) for figure 2:

**Figure supplement 1.** Comparison between simulated and experimental data.

background increased the empirical lifetime, as they are evenly distributed across lifetime time channels and have a higher mean lifetime than FLIM-AKAR (*Figure 2*; *Figure 2B*: p<0.0001, final simulated data vs sensor + autoF for all sensor photon number conditions). Afterpulse and background had a greater impact on empirical lifetime compared with autofluorescence. This is because the empirical lifetime of background/afterpulse (~4.90 ns) deviates more from FLIM-AKAR (2–2.3 ns) than the empirical lifetime of autofluorescence (around 1.69 ns) does. Thus, while autofluorescence affects both the fitted $P_1$ and empirical lifetime, only empirical lifetime is sensitive to afterpulse and background.

Importantly, the biases introduced by autofluorescence, afterpulse, and background were less pronounced at higher sensor photon counts, which can be explained by the relatively small contribution of these factors when sensor fluorescence is high. Notably, as the number of sensor photons increased, the biases not only decreased but also plateaued, which made fluorescence lifetime less dependent on the number of sensor photons (*Figure 2*). Furthermore, the variance also decreased at higher photon counts for both the fitted $P_1$ and empirical lifetime (*Figure 2B*). Thus, higher photon counts resulted in both less bias and less noise.

To quantify how well our simulation matches the experimental conditions, we calculated fitted $P_1$ and empirical lifetime after each simulation step, with $P_1$ matching the experimental condition. For both fitted $P_1$ and empirical lifetime, the values from final simulated data were not significantly different from those from experimental data (*Figure 2—figure supplement 1C*; adjusted p=0.81 for empirical lifetime and 0.98 for fitted $P_1$, final simulated data vs experimental data; n=500 and 7, respectively). Thus, our simulated fluorescence lifetime data recapitulated experimental data in biologically relevant settings.

In summary, using FLiSimBA, we realistically simulated fluorescence lifetime data in biological settings and quantitatively defined how different sources of signals contribute to bias and noise. These curves and quantitative approaches can, therefore, be used to evaluate the impact of optimization of these different sources on fluorescence lifetime measurements.

## Determination of minimum photon number requirements to achieve specific SNRs

How many photons do we need for a given FLIM experiment in a biology experiment? Are more photons always better? A greater number of photons translates into a better SNR but also a lower sampling rate and reduced size for the field of view. How can we quantitatively find the optimal compromise among these factors? As sensor fluorescence increases, the variances of both the fitted $P_1$ and empirical lifetime decrease (*Figure 2B*), and the ability to detect a specific amount of fluorescence lifetime response increases. Although the number of photons required to achieve a certain SNR was analyzed previously (*Esposito et al., 2007*; *Esposito et al., 2008*; *Gerritsen et al., 2002*; *Köllner and Wolfrum, 1992*; *Kumar, 2012*; *Lakowicz, 2006*; *Nasser and Meller, 2022*; *Netaev et al., 2022*;

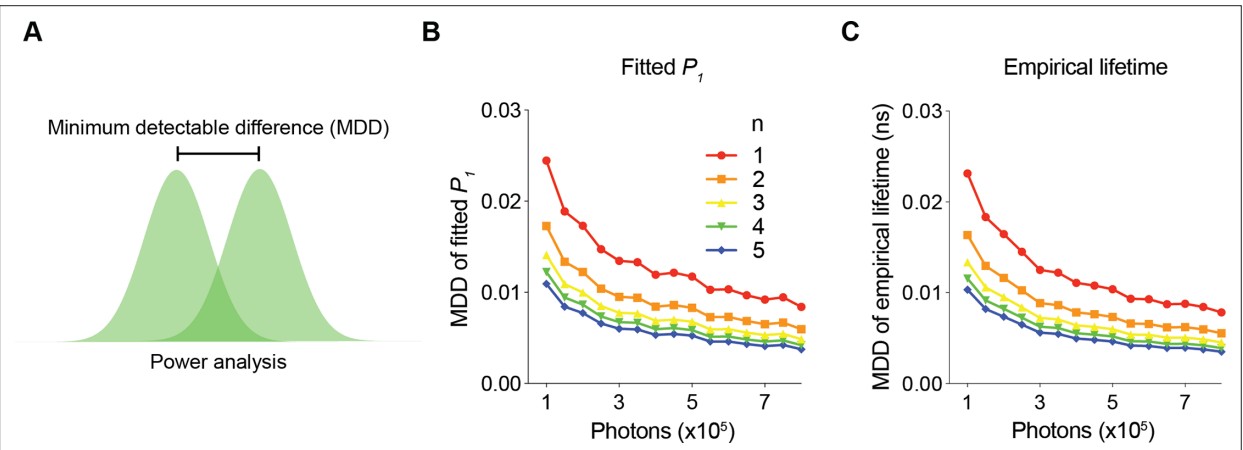

**Figure 3.** The minimum number of photons required to detect a specific fluorescence lifetime response. (**A**) Schematic illustrating the power analysis used to calculate the minimum detectable difference between two lifetime distributions. (**B, C**) Minimum detectable differences in the fitted $P_1$ (**B**) and empirical lifetime (**C**) for different numbers of sensor photons and for different numbers of pairs of sampled data (n). The data were simulated with $P_1$=0.5.

*Philip and Carlsson, 2003*; *Roethlein et al., 2015*; *Trinh and Esposito, 2021*; *Turton et al., 2003*; *Walsh et al., 2016*; *Xiao et al., 2021*; *Yasuda, 2006a*), such analysis was not performed with consideration of biological samples in realistic experiments involving autofluorescence, background, and afterpulse.

To determine the minimum number of photons required for a certain SNR, we analyzed the minimum detectable differences (MDD) in both the fitted $P_1$ and empirical lifetime for different numbers of sensor photons and repeated measurements (*Figure 3*). The MDD was calculated with 80% power and a 5% significance level. As the number of sensor photons increased, the MDDs decreased. As the number of repeated measurements increased, the MDD also decreased (*Figure 3B and C*). Importantly, the MDD curves provide quantitative information on the minimum number of photons required for a certain amount of expected signal. For example, to detect a $P_1$ change of 0.006 or a lifetime change of 5 ps with five sample measurements in each comparison group, approximately 300,000 photons are needed. As the sensor fluorescence increased, the gain in SNR decreased. Thus, MDD curves generated with FLiSimBA are instrumental for determining the optimal experimental conditions (for example, optimal imaging speed and sizes of imaging fields) necessary to detect a specific amount of lifetime change.

Using realistic determination of the SNR with FLiSimBA, we also quantitatively evaluated the impact of specific hardware changes on the SNR. HBDs are advantageous over traditional gallium arsenide phosphide (GaAsP) PMTs because of their narrower IRF width and lack of afterpulsing (*Becker et al., 2011*). However, hybrid detectors are more expensive. Thus, it is valuable to quantify the benefits of HBDs over traditional GaAsP PMTs for SNRs. We generated fluorescence lifetime histograms with narrower IRFs and no afterpulses for HBDs and compared them with the simulated data for traditional GaAsP PMTs (*Figure 4A and B*). Hybrid detectors and GaAsP PMTs displayed similar photon-dependent changes in fitted $P_1$ and empirical lifetime (*Figure 4*). Furthermore, a given sensor photon number gave a comparable MDD for HBDs and GaAsP PMTs (*Figure 4*). The similarity in the MDD curves can be explained by similar levels of variance between HBDs and PMTs (*Figure 4C and D*): although HBDs with a narrower IRF introduced less noise than GaAsP detectors (*Figure 4G and H*; noise from sampling from the IRF: standard deviation (STD) of fitted $P_1$: 0.0008 for GaAsP PMTs and 0.0005 for HBDs; STD of empirical lifetime: 0.0003 for GaAsP PMTs and 0.0001 for HBDs), the noise of lifetime measurements was dominated by photon noise, and not by IRF noise (*Figure 4G and H*; noise from sampling of photons: STD of fitted $P_1$: 0.0028 for both GaAsP PMTs and HBDs; STD of empirical lifetime: 0.0025 for both GaAsP PMTs and HBDs). Thus, although HBD can offer other advantages (*Becker et al., 2011*; *Trinh and Esposito, 2021*), HBDs with a narrower IRF and no afterpulse yield little improvement in the SNR for fluorescent protein-based sensors in biological applications.

In summary, the quantitative and biologically realistic curves generated with FLiSimBA allow users to determine the appropriate trade-off in each experiment. Given the specific sensor brightness, sensor lifetime, and expected signal amplitude in specific biological applications, FLiSimBA allows the users to select the imaging speed and size of the imaging field to achieve the desired SNR. In addition, FLiSimBA allows users to evaluate the trade-off between performance improvement and price with specific hardware changes.

## Expression level dependence of fluorescence lifetime estimates

Sensor expression levels often change over days and across animals and is usually assumed not to influence lifetime estimates because fluorescence lifetime is an intensive property of fluorophores. However, this assumption is true only if the biosensor is the only contributor to measurements. With autofluorescence, afterpulse, and background signals present in biological applications, the amount of sensor fluorescence relative to these contributing factors can lead to an apparent change in fluorescence lifetime estimates even if the biosensor is in the same conformational state (*Ma et al., 2024*). Here, we challenge the conventional view that fluorescence lifetime is independent of sensor expression and use simulation to define the range in which sensor expression has a negligible influence on lifetime.

We first determined how sensor expression levels altered the relative responses in fitted $P_1$ and empirical lifetime (*Figure 5A*). When $P_1$ changed from 0.4 to 0.5, both fitted $P_1$ and empirical lifetime increased as expected (*Figure 5B and C*; adjusted p<0.0001, $P_1$=0.4 vs $P_1$=0.5, under all photon number conditions for both fitted $P_1$ and empirical lifetime). As the sensor photon count increased,

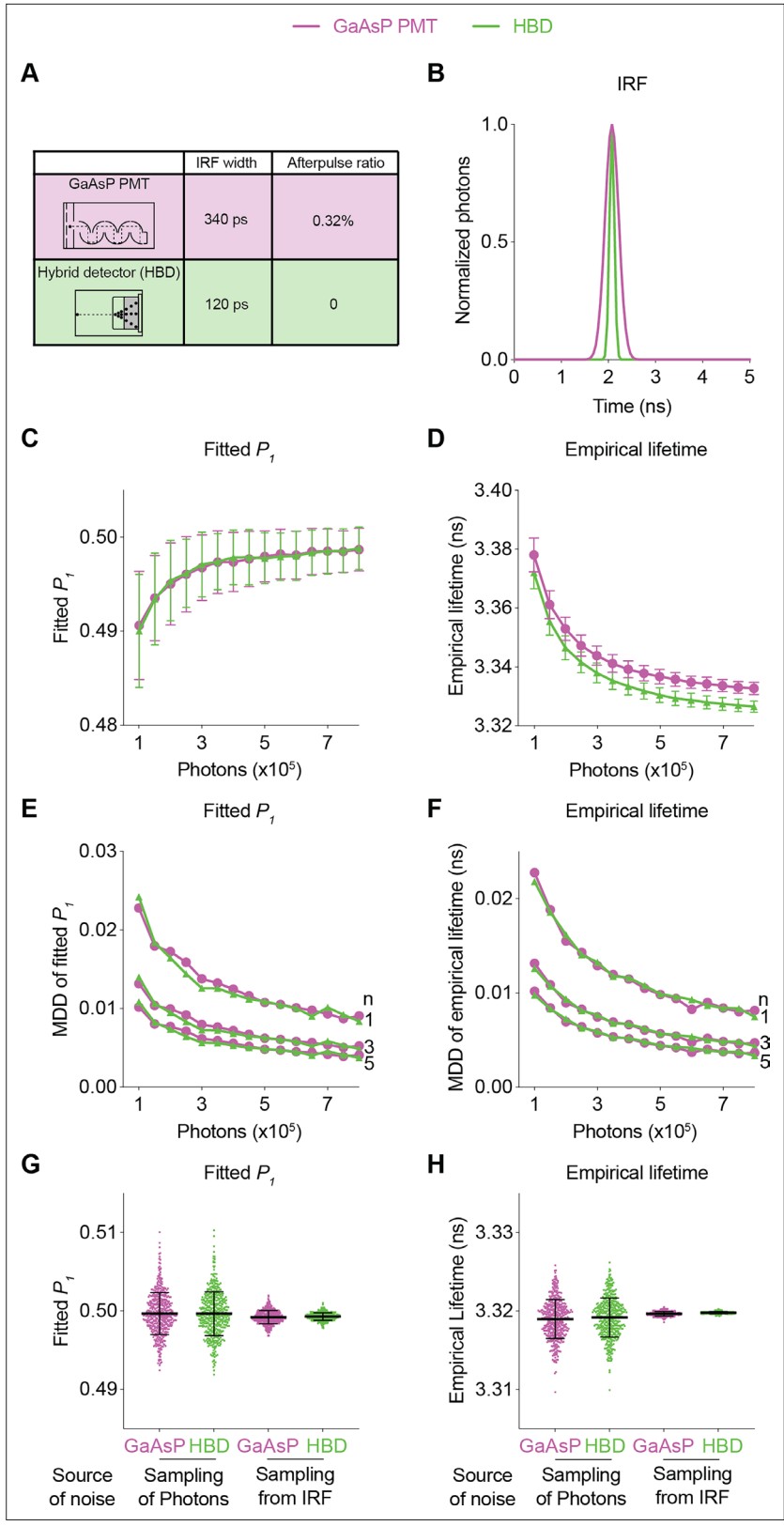

**Figure 4.** Comparison of the fluorescence lifetime responses of GaAsP photomultiplier tubes (PMTs) and hybrid detectors (HBDs). (**A**) Differences in instrument response function (IRF) widths and afterpulse ratios between the GaAsP PMT and HBD. These specific parameters are used in subsequent simulations. (**B**) Gaussian IRFs used for simulation, reflecting different Gaussian widths for the GaAsP PMT and HBD. (**C, D**) Distributions of the fitted

*Figure 4 continued on next page*

*Figure 4 continued*

$P_1$ (**C**) and empirical lifetime (**D**) of the simulated data from the GaAsP PMT or HBD, with simulated $P_1$=0.5 and showing the sensor photon number dependence. The data are represented as means and standard deviations. (**E, F**) Minimum detectable difference in the fitted $P_1$ (**E**) and empirical lifetime (**F**) for different numbers of sensor photons and different numbers of data samples. The data were simulated with $P_1$=0.5. (**G, H**) Distribution of the fitted $P_1$ (**G**) and empirical lifetime (**H**) of the simulated data from the GaAsP PMT or HBD, with noise introduced from either sampling of photons or sampling from the IRF distribution. The data were simulated without autofluorescence, afterpulse, or background and are represented as means and standard deviations. n=500 simulations.

---

fitted $P_1$ increased and empirical lifetime decreased with both plateauing at high photon counts (*Figure 5B and C*; p<0.05, 800,000 photons vs 550,000 photons or less for fitted $P_1$, 800,000 photons vs 700,000 photons or less for empirical lifetime, for both $P_1$=0.4 and $P_1$=0.5). This photon-dependent change in fitted $P_1$ or lifetime is consistent with the diminishing effect of autofluorescence, background, and afterpulse. We next determined whether the response amplitude also changed as the sensor photon count increased by comparing the changes in fitted $P_1$ and empirical lifetime when actual $P_1$ changed from 0.4 to 0.5. As the sensor photon count increased, the change in the fitted $P_1$ and empirical lifetime also showed an apparent increase (*Figure 5D and E*; p<0.05, 800,000 photons vs 450,000 photons or less for fitted $P_1$ and 400,000 photons or less for empirical lifetime). The response dependence on photons approached asymptotes at high photon counts: With the parameters used in our case study, the fluorescence lifetime responses were relatively stable at sensor photon counts of 500,000 and above (*Figure 5D and E*). Thus, our results challenge the widely held view that relative fluorescence lifetime changes are independent of sensor expression level and define a quantitative threshold of sensor expression that allows comparison of lifetime responses across different sensor expression levels.

Since fluorescence lifetime allows comparison of absolute levels of analytes (*Brinks et al., 2015*; *Lazzari-Dean et al., 2019*; *Ma et al., 2024*; *van der Linden et al., 2021*; *Zheng et al., 2015*), we determined the range where changes in sensor expression do not significantly affect fitted $P_1$ or empirical lifetime (*Figure 6A*). A higher sensor photon count reduces the likelihood of significant changes in lifetime due to sensor expression (*Figure 6A*). Thus, we determined the minimum number of sensor photons needed for a specific change in sensor expression to have a negligible impact on lifetime measurements (*Figure 6A*). We first determined the apparent change in fitted $P_1$ and empirical lifetime introduced by the change in sensor expression levels (*Figure 6C*, purple traces). As sensor fluorescence increased, the sensor expression-induced apparent change in fitted $P_1$ and empirical lifetime decreased. Additionally, by considering the variance associated with a specific photon count, we calculated the minimum change in fitted $P_1$ or empirical lifetime that would result in a statistically significant difference due to sensor expression changes (*Figure 6C*, black traces, t test threshold, see Materials and methods). The intersection of these two traces (*Figure 6C*) indicates the minimum number of sensor photons required to tolerate a certain amount of sensor photon changes (i.e. t tests did not show a statistically significant difference) (*Figure 6B*). As the total number of sensor photons increased, a greater difference in sensor expression was tolerated. Fitted $P_1$ could tolerate more photon changes than empirical lifetime for a given number of sensor photons (*Figure 6B*). These curves provide quantitative guidelines for determining the necessary sensor photon counts to compare absolute lifetime values across different sensor expression levels.

Together, the data in *Figures 5 and 6* not only challenge the conventional view that fluorescence lifetime is independent of sensor expression levels in biology experiments, but also, more importantly, quantitatively define the range of sensor expression levels that enable comparison of either relative responses or absolute lifetime values across diverse sensor expression levels. These results are critical for guiding experiments that involve comparisons across animals, across brain regions, and across chronic time periods.

## Innovating to reduce sensor expression dependence: avoiding autofluorescence and lowering background signal

Sensor expression dependence limits comparisons of fluorescence lifetime values and responses across chronic periods and across animals. Thus, we investigated the extent to which the dependence

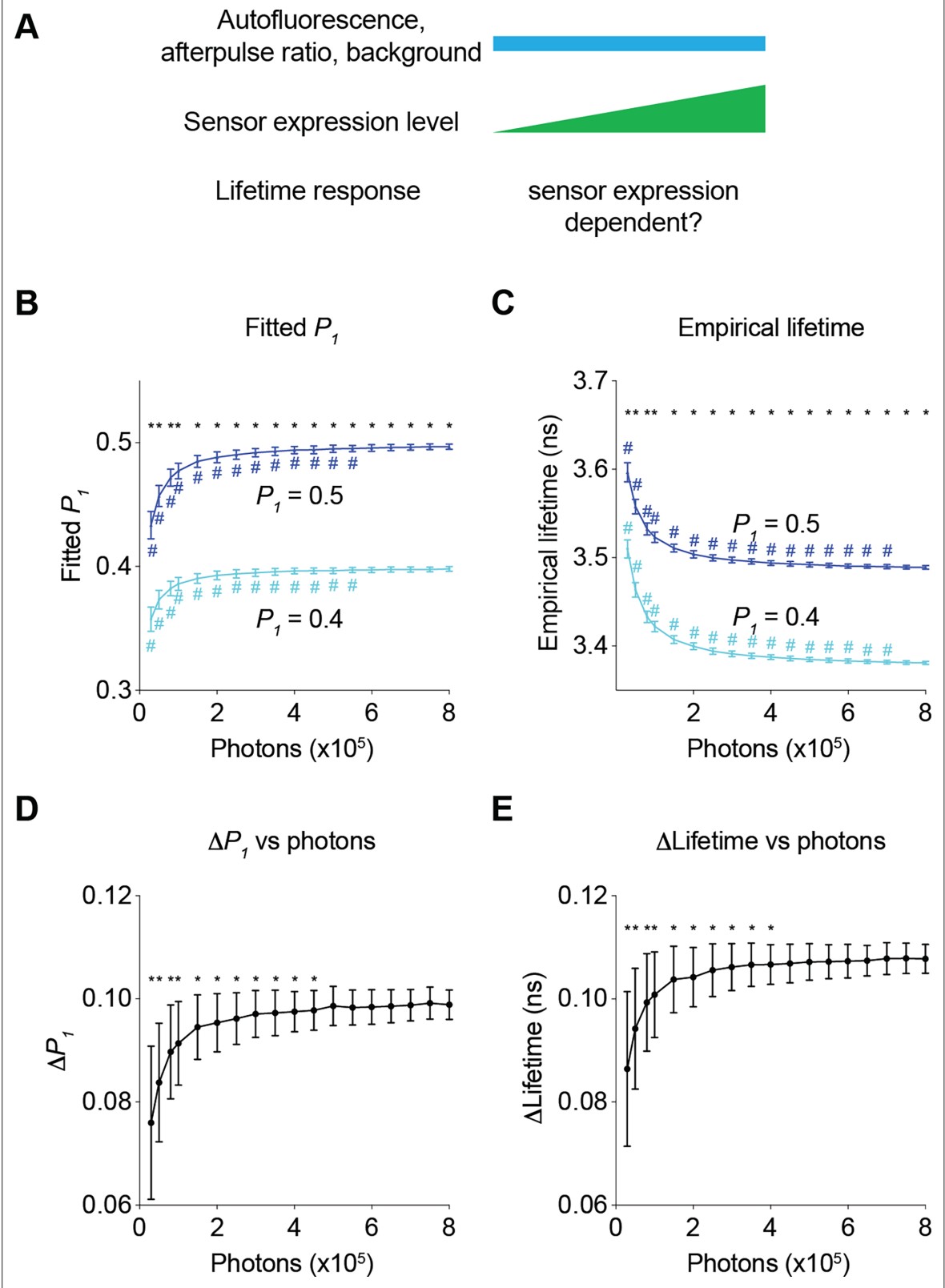

**Figure 5.** Impact of the number of sensor photons on the response amplitudes of the fitted $P_1$ and the empirical lifetime. (**A**) Schematic illustrating the following question under investigation: in biological systems, as autofluorescence, the afterpulse ratio, and the background remain constant, does the lifetime response change as sensor brightness increases? (**B, C**) Distribution of the fitted $P_1$ (**B**) and empirical lifetime (**C**) from fluorescence lifetime data with simulated $P_1$ values of 0.4 and 0.5 across different sensor photon numbers. *$p<0.05$, $P_1$=0.4 vs $P_1$=0.5, two-way ANOVA with Šídák's multiple

*Figure 5 continued on next page*

*Figure 5 continued*

comparisons test; #p<0.05, vs photons = 800,000, one-way ANOVA with Dunnett's multiple comparisons test. (**D, E**) Distribution of the changes in the fitted $P_1$ (**D**) and empirical lifetime (**E**) for different sensor photon numbers. The simulated $P_1$ varied from 0.4 to 0.5. *p<0.05, vs photon count = 800,000, one-way ANOVA with Dunnett's multiple comparisons test. The data are represented as means and standard deviations. n=500 simulations.

of sensor expression could be mitigated by achievable technological advances, in particular, by developing sensors outside the autofluorescence spectra (e.g. red-shifted sensors), and/or by lowering background signal (e.g. with photodetectors with low dark current, together with better light proofing) (*Figure 7*).

If sensors could be made with emission spectra that do not overlap with tissue autofluorescence (e.g. red or far-red sensors), both the relative response and absolute values of fitted $P_1$ no longer depended on sensor expression (*Figure 7A and C* left panels; 7A: p=0.047 for 30,000 photons, p>0.8 for the other photon counts, vs 800,000 photons). This expression independence can be explained

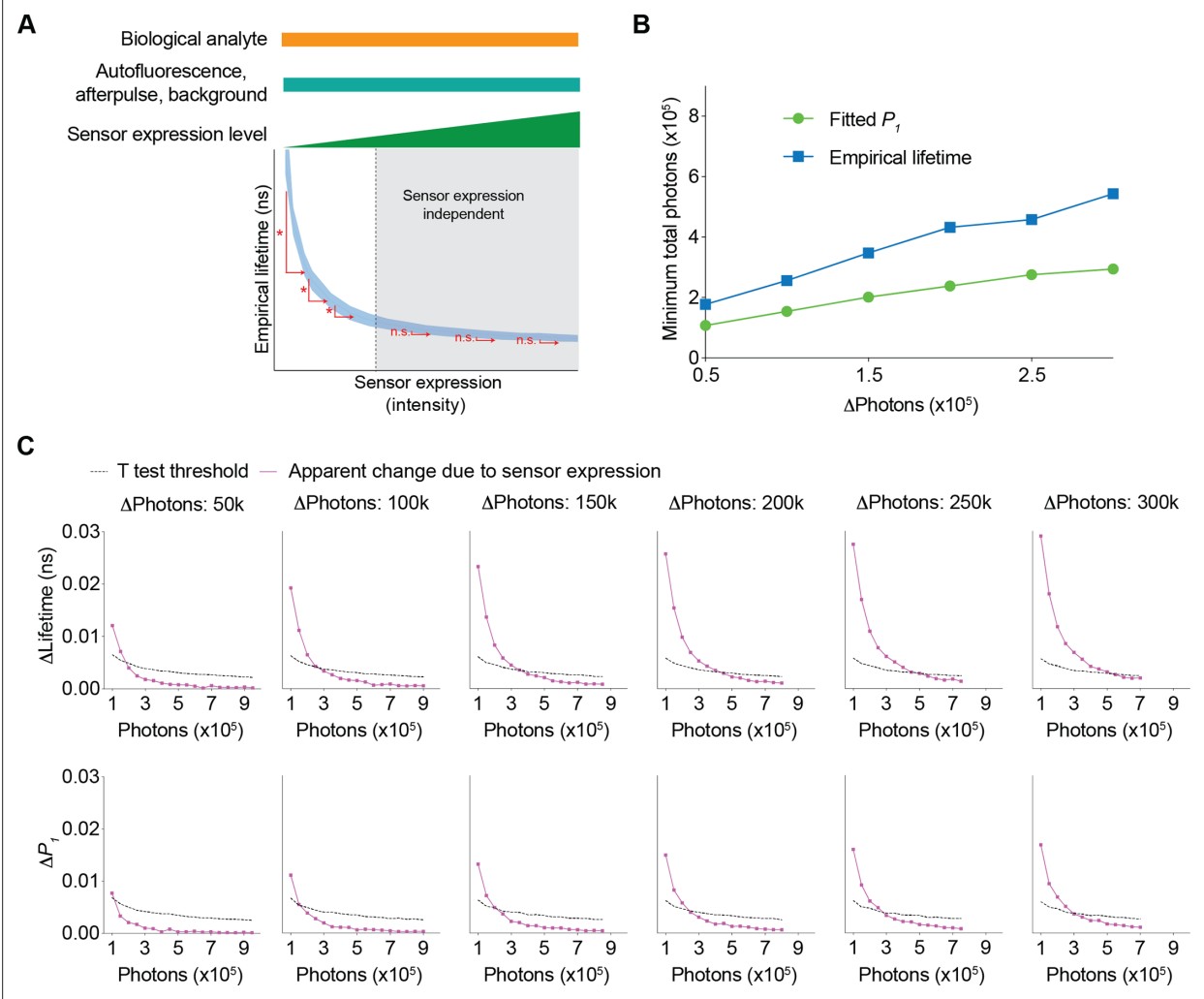

**Figure 6.** Conditions needed for the sensor expression-induced lifetime change to be statistically nonsignificant. (**A**) Schematic illustrating the question under investigation: as the biological analyte, autofluorescence, afterpulse ratio, and background signals remain constant, as the sensor expression/brightness increases, what is the minimum number of sensor photons needed to tolerate a specific change in sensor expression such that the apparent lifetimes are not significantly different? (**B**) Relationship between changes in photon number due to changes in expression level and the minimum number of sensor photons required not to reach statistical significance for both the fitted $P_1$ and empirical lifetime. The data were simulated with $P_1$=0.5. The minimum number of sensor photons was calculated by interpolating the intersection between the two curves in (**C**). (**C**) Plots of changes reaching statistical significance according to t tests (calculated as 1.96*standard error of the difference in the mean) and apparent changes due to sensor expression for both the empirical lifetime (upper panels) and fitted $P_1$ (lower panels). The data were simulated with $P_1$=0.5. Different sensor expression changes are plotted in each panel. The data are represented as means and standard deviations.

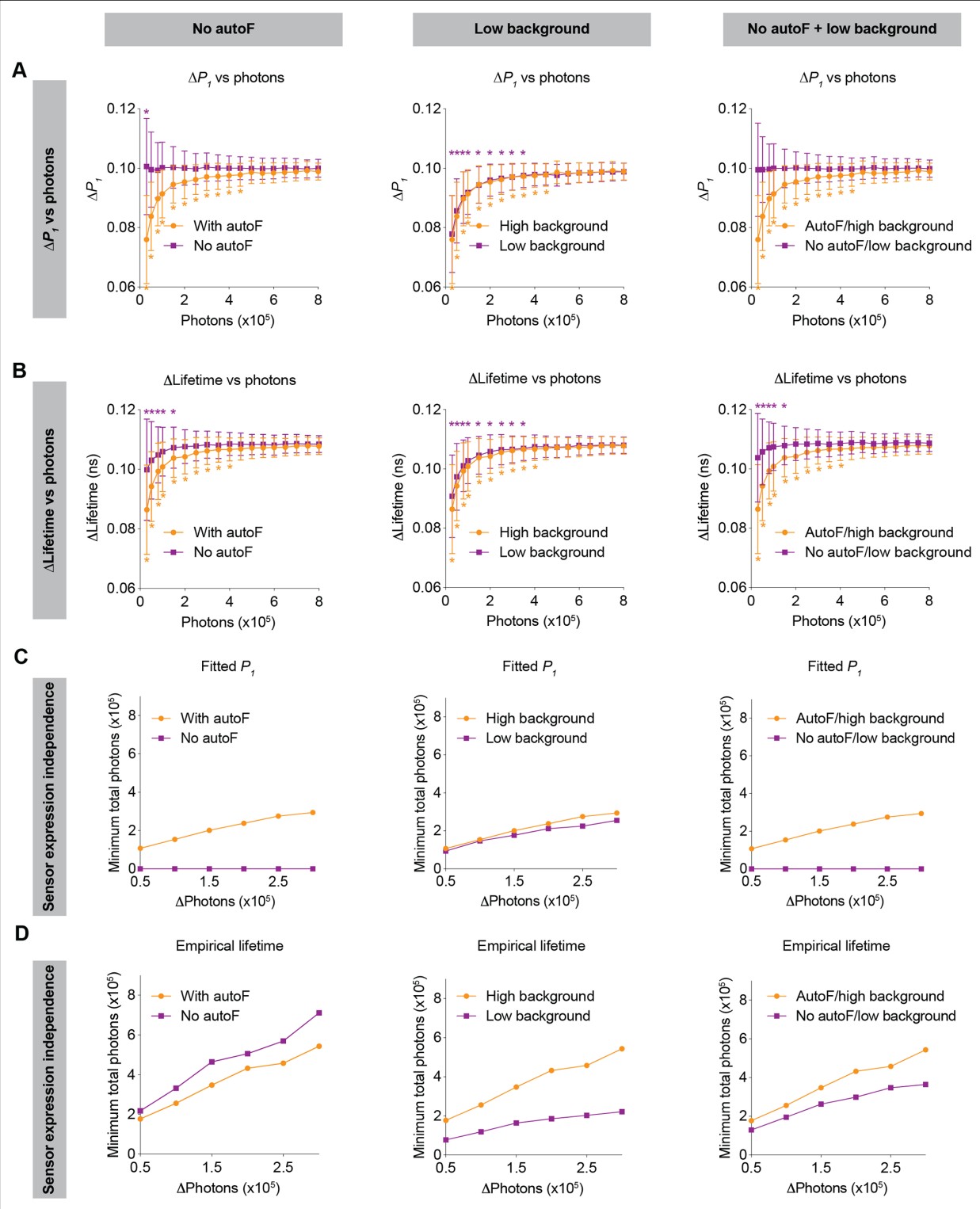

**Figure 7.** Impact of no autofluorescence and/or low background on the lifetime dependence of sensor expression. (**A, B**) Distribution of the change in fitted $P_1$ (**A**) and empirical lifetime (**B**) with different sensor photon numbers, under no autofluorescence, low background, and both improvement conditions. The simulated $P_1$ varied from 0.4 to 0.5. *$p<0.05$, vs photon count = 800,000, one-way ANOVA with Dunnett's multiple comparisons test. n=500 simulations. The data are represented as means and standard deviations. (**C, D**) Relationships between changes in photon number due to expression level and the minimum number of sensor photons required not to reach statistical significance for both the fitted $P_1$ (**C**) and empirical lifetime (**D**) under no autofluorescence, low background, or both improvement conditions. The data were simulated with $P_1$=0.5. The orange traces are the same as the corresponding data in *Figures 5 and 6* and were reused here for comparison purposes.

by the fact that only autofluorescence contributes to the sensor expression-dependent bias of fitted $P_1$ (**Figure 2**). With no autofluorescence, the relative response in empirical lifetime showed less dependence on sensor expression (**Figure 7B**, left, $p<0.05$ for 150,000 photons or less, $p>0.2$ for the other photon counts, vs 800,000 photons), whereas the absolute empirical lifetime could tolerate less change in sensor expression (**Figure 7D**, left). Thus, sensors with spectra that avoid overlap with autofluorescence are completely independent of sensor expression levels when fitted $P_1$ is used as the metric of evaluation.

When background signals were lower, both the relative response and absolute values of fitted $P_1$ showed a similar sensor expression dependence to that of the higher background (**Figure 7A and C**, middle panels; 7A: $p<0.05$ for 350,000 photons or less, $p>0.1$ for the other photon counts, vs 800,000 photons). This is because the background is considered in the fitting process as a mathematical term. For the empirical lifetime response, a lower background slightly reduced the sensor expression dependence (**Figure 7B**, middle, $p<0.05$ for 350,000 photons or less, $p>0.1$ for the other photon counts, vs 800,000 photons). Notably, a lower background signal caused greater tolerance of sensor expression level changes for the measurement of absolute empirical lifetime (**Figure 7D**, middle). Thus, a lower background does not significantly alter the sensor level dependence when fitted $P_1$ is used as the metric of measurement, but it significantly reduces the sensor level dependence when empirical lifetime is used as the metric of measurement.

Consistent with the results above, when there was no autofluorescence and the background was lower, there was no longer sensor expression dependence for fitted $P_1$ (**Figure 7A and C**, right panels; 7A: $p>0.99$ for all photon counts vs 800,000 photons), and the sensor expression dependence also became much less pronounced for empirical lifetime (**Figure 7B and D**, right; 7B: $p<0.05$ for 150,000 photons or less, $p>0.3$ for the other photon counts, vs 800,000 photons).

In summary, using FLiSimBA, we show quantitatively the benefit gained from future technological innovations: Developing sensors outside autofluorescence spectra or having a lower background can significantly reduce the dependence of fluorescence lifetime on sensor expression in biological systems, increasing the versatility of lifetime comparison across animals, brain regions, and chronic time scales, as well as allowing us to quantify the absolute levels of analytes.

## Innovating for multiplexing: revealing the feasibility and specifying sensor characteristics for multiplexed imaging with both fluorescence lifetime and intensity

Understanding of any biological systems requires the quantification of the dynamics of many biological signals that interact together. Multiplexed imaging with sensors with different excitation and emission spectra is powerful, but only a few color channels can be effectively separated. Combining fluorescence lifetime and intensity could greatly enhance the capacity to monitor more biological signals simultaneously, and this idea is becoming increasingly feasible with the development of genetically encoded and chemigenetic sensors (**Farrants et al., 2024**; **Frei et al., 2022**; **van der Linden et al., 2021**). However, this approach was only used to distinguish sensors with different lifetimes in different spatial compartments (**Frei et al., 2022**; **Rahim et al., 2022**), or fluorophores that do not change dynamically and thus do not act as sensors (**Hamilton and Sanabria, 2019**; **Rahim et al., 2022**; **Rice and Kumar, 2014**; **Widengren et al., 2006**), and there was no quantitative framework for exploring the multiplexing capacity to analyze the dynamics of biosensors. Here, we investigated whether the combined signal of sensors with different lifetimes can be effectively deconvolved to quantitate the dynamic intensity change of each sensor – if this is feasible, this new approach could be used to simultaneously track the dynamics of multiple signals in the same spatial location, or in photometry experiments (**Figure 8A**), thus greatly enhancing our ability to understand the systems biology of signaling.

We modeled a multiplexed system with two sensors, each displaying a single exponential decay in its fluorescence lifetime histogram but with different decay constants. Furthermore, in response to a change in the concentration of a biological analyte, the fluorescence lifetime of each sensor remains constant, whereas the fluorescence intensity is modulated. We used FLiSimBA to simulate multiplexed fluorescence lifetime imaging with realistic parameters and noise in biological systems. Subsequently, we deconvolved the combined histogram by fitting it with a double exponential decay equation (**Equation 3**, Gauss-Newton nonlinear least-square fitting algorithm), whose decay time constants corresponded to those of the two sensor species (**Figure 8A**).

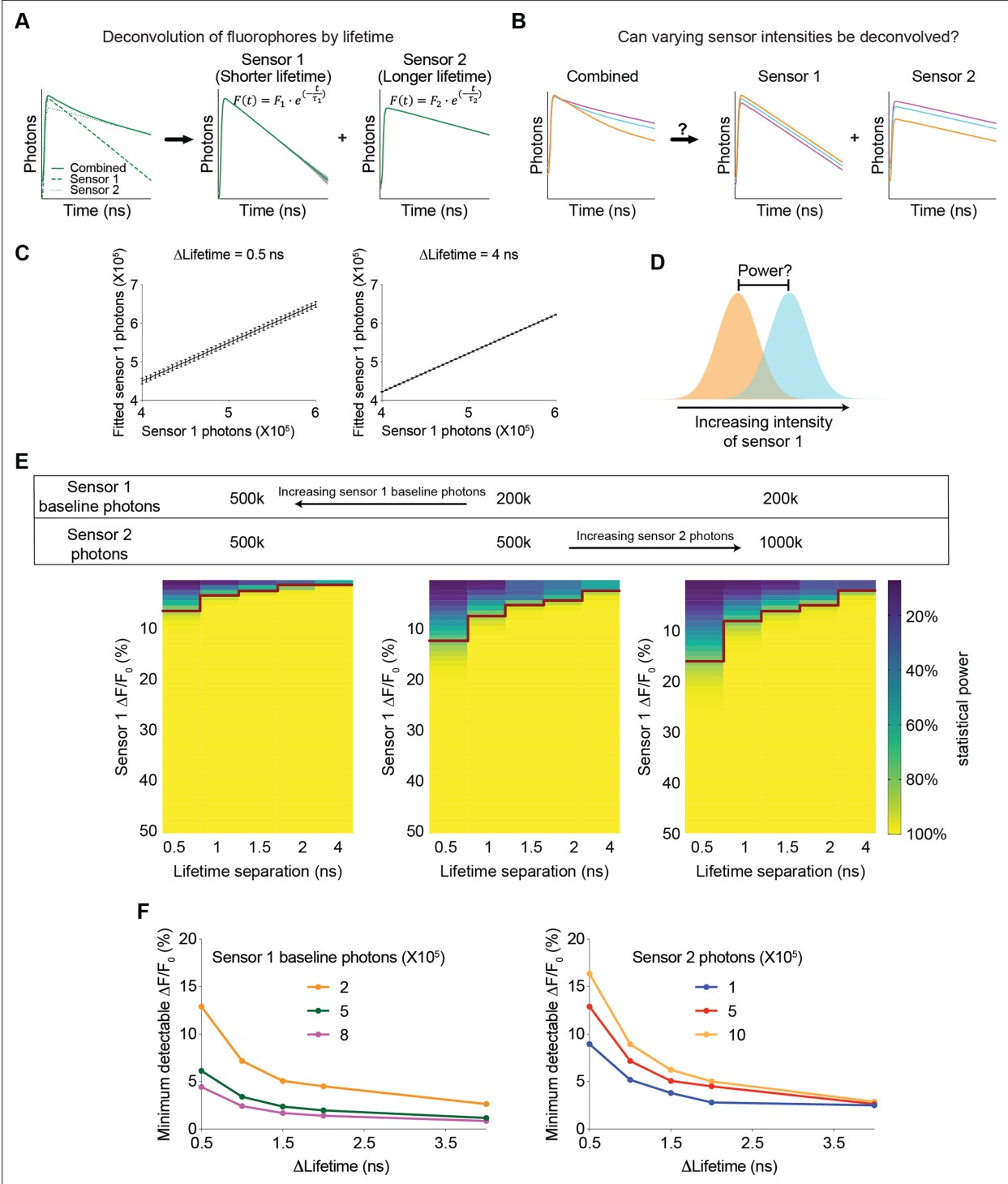

**Figure 8.** Feasibility and parameter requirements for multiplexed fluorescence intensity and lifetime imaging. (**A**) Schematic illustrating the motivation for the question. Sensor 1 and sensor 2 are intensity-based sensors with the same color but different fluorescence lifetimes (with single exponential decay constants $\tau_1$ and $\tau_2$, respectively). The fluorescence signals are combined, and then mathematically deconvolved into individual components. Dotted lines: true fluorescence lifetime histograms of the two sensors; solid lines: simulated fluorescence lifetime histograms with standard deviation (gray shading) before and after deconvolution. (**B**) Schematic illustrating the question under investigation: when sensor 1 and sensor 2 change fluorescence intensities and their combined fluorescence is collected simultaneously, can the respective intensities be deconvoluted from the combined measurements? (**C**) Deconvoluted number of sensor 1 photons based on different simulated sensor 1 photons. Sensor 2 was simulated with a photon count of 500 k. The left and right graphs are based on either 0.5 or 4 ns differences in the single exponential decay constants between the two sensors. The left and right graphs show that the variance in deconvoluted sensor 1 photons is less when the lifetime separation between sensor 1 and sensor 2 is

*Figure 8 continued on next page*

*Figure 8 continued*

greater. n=500 simulations. The data are represented as means and standard deviations. (**D**) Schematic illustrating the calculation of the statistical power for detecting a specific change in sensor 1 photon counts. (**E**) Heatmaps showing the statistical power of detecting specific changes in the intensity of sensor 1 with different sensor 1 baseline photons, different sensor 2 photons, and different lifetime separations between the two sensors. The red line denotes the change in intensity that provides a statistical power of 80%. (**F**) Relationship between the minimum detectable intensity change in sensor 1 to reach 80% statistical power and the lifetime difference between the two sensors. Curves were plotted with different sensor 1 baseline photons (left, 500 k sensor 2 photons) and different sensor 2 photons (right, 200 k sensor 1 baseline photons).

The online version of this article includes the following figure supplement(s) for figure 8:

**Figure supplement 1.** Minimum detectable intensity change across diverse experimental conditions for multiplexed fluorescence lifetime imaging.

We first investigated whether varying sensor intensities could be accurately deconvolved (*Figure 8B*). Deconvolution accurately recovered the photon numbers attributed to each sensor compared with the actual sensor photons inputted into the model (*Figure 8C*). The number of fitted sensor 1 photons (with shorter lifetime) was greater than the number of simulated sensor 1 photons, consistent with the contribution of autofluorescence whose average lifetime is closer to that of Sensor 1. Furthermore, when the lifetime decay constants of the two species were further separated, the variance in the fitted sensor photons decreased (*Figure 8C*, STD = 0.012 for 4 ns separation and STD = 0.053–0.061 for 0.5 ns separation), indicating that further separation of lifetimes between the two sensors resulted in a better SNR.

To quantitatively evaluate the feasibility of multiplexed fluorescence lifetime imaging of sensors with the same spectra, we calculated the statistical power for detecting specific intensity changes of sensor 1 photons (*Figure 8D*). The statistical power for detecting a specific intensity change was greater when the intensity change was greater and when the difference between the lifetime constants of the two sensors was greater (*Figure 8E*, for any given heatmap). Furthermore, the statistical power was also greater when the number of starting sensor 1 photons was greater and the number of sensor 2 photons were lower (*Figure 8E*, comparing between the heatmaps). Therefore, these results were valuable for demonstrating the feasibility of using fluorescence lifetime for multiplexed imaging and for quantitatively specifying the parameter space that provided sufficient statistical power.

To systematically define the parameter space, we determined the minimum intensity change that was detectable under different conditions. For any given sensor 1 starting photon count, sensor 2 photon count, and lifetime separation, we fitted the statistical power versus intensity change relationship to a 5-parameter logistic curve (*Figure 8—figure supplement 1A*). Using the fitted curve, we calculated the intensity change that would yield a statistical power of 80% and defined it as the minimum detectable $\Delta F/F_0$ (*Figure 8—figure supplement 1A*). The minimum detectable $\Delta F/F_0$ for sensor 1 decreased as the lifetime separation increased, as sensor 1 expression level increased, or as the number of sensor 2 photons decreased (*Figure 8F* and *Figure 8—figure supplement 1B*). If the two sensors had comparable dynamic ranges and comparable expression levels, the minimum detectable $\Delta F/F_0$ for sensor 2 was similar to that of sensor 1 (*Figure 8—figure supplement 1C*). Remarkably, only a 0.5 ns separation in lifetime constants between the two sensors could result in a minimum detectable $\Delta F/F_0$ of 5% in sensor 1 if sensor 1 showed high but reasonable expression (800,000 photons or higher) and sensor 2 showed reasonable expression (500,000 photons or lower) (*Figure 8F*). These results showed how the sensitivity of the intensity response quantitatively depended on the sensor expression and differences in lifetime between the two sensors.

In summary, our simulation showed that multiplexed fluorescence lifetime imaging was feasible with small shifts in the sensor properties. Furthermore, FLiSimBA could be used to quantitatively define the parameters of the sensors to yield a desired sensitivity. Thus, these results provide sensor developers with specific sensor properties to evolve toward. With simultaneous spectral and lifetime multiplexing, our proposed method enables the detection of a much larger number of signals dynamically, opening doors to better systems understanding of signaling within the cell and physiological interactions between different organs.

## Discussion

Here, we provide a quantitative framework for analyzing fluorescence lifetime in realistic biological settings. We introduce FLiSimBA, a platform that accurately simulates fluorescence lifetime data for

biological applications (*Figures 1 and 2*). With FLiSimBA, we address key questions in FLIM. First, to understand SNRs, we determined the number of photons required for different minimum detectable differences (*Figure 3*). Additionally, we assessed the impact of hardware changes on SNRs by comparing GaAsP PMTs with hybrid detectors (*Figure 4*). Moreover, we challenged the conventional view that biosensor expression levels do not affect fluorescence lifetime. We revealed how variation in sensor expression influences response amplitude (*Figure 5*) and identified the amount of expression level variation that does not significantly alter fluorescence lifetime estimates (*Figure 6*). Finally, we evaluated the feasibility, quantified the benefits, and specified the parameter space for FLIM innovations that minimize expression level dependence (*Figure 7*) or enable multiplexed imaging with lifetime and intensity (*Figure 8*). In summary, our study provides valuable insights and a quantitative framework for defining the power and limitations of fluorescence lifetime experiments in biological applications.

## FLiSimBA as a tool for realistic simulation of FLIM and FLiP experiments

FLiSimBA is a necessary and useful tool for FLIM and FLiP experiments for a few reasons. First, it realistically simulates lifetime data from biological applications. Second, FLiSimBA and the quantitative framework we provide can precisely define the benefits and limitations of FLIM or FLiP experiments, enabling rigorous experimental design and data interpretation. Third, the versatility of FLiSimBA allows easy adaption to different sensors, tissues, organisms, and analysis methods (*Bajzer et al., 1991*; *Chen et al., 2022*; *Jo et al., 2005*; *Mannam et al., 2020*; *Periasamy, 1988*; *Rowley et al., 2011*; *Shen et al., 2024*; *Smith et al., 2019*; *Steinbach, 2012*; *Stringari et al., 2011*; *Wang et al., 2022*; *Wu et al., 2016*; *Xiao et al., 2023*; *Zickus et al., 2020*).

While this study used specific parameters from FLIM-AKAR and our imaging systems to demonstrate the power of FLiSimBA, FLiSimBA allows customizable inputs (*Figure 1*) with minimal model assumptions to suit specific experimental settings. Users should carefully select mathematical models for their fluorescence sensors (*Chen et al., 2014*; *Steinbach, 2012*), measure autofluorescence that matches the age, brain region, and data collection conditions (e.g. ex vivo or in vivo) of the tissue (*Campbell et al., 2024*; *Jang et al., 2022*; *Morrow et al., 2024*; *Tehrani et al., 2023*), as well as measure afterpulse ratio, background, and dark currents relevant to their specific FLIM or FLiP settings. While these input parameters do not alter the general insights from this study, they affect quantitative values and it is thus crucial to accurately match experimental conditions to enhance the rigor of experimental design and data interpretation.

## Insights into measurement uncertainty of FLIM

Beyond providing a package to realistically simulate FLIM data, our results also provide insights into critical questions for FLIM usage in biology that were not previously available. Quantifying the uncertainty of fluorescence lifetime measurements is a key aspect of any measurement but is often neglected. Our results provide the amplitude of the error bars and reveal the sources of bias and noise in fluorescence lifetime measurements in realistic biological experiments. Therefore, these results facilitate the disambiguation of signals from noise in data interpretation and empower FLIM users to design optimal experiments by precisely evaluating the compromise between the SNR, field-of-view size, and imaging speed.

## Insights into expression level dependence of FLIM

Our analysis of expression level dependence demonstrates that small quantitative changes can impact qualitative conclusions: Because of the influence of autofluorescence, afterpulse, and background, changes in sensor expression levels can lead to changes in lifetime measurements. This challenges the widely held assumptions about the very advantage of fluorescence lifetime measurements: Its insensitivity to sensor expression levels. While this issue is less problematic when the same sample is compared over short periods (for example, minutes), it can lead to misinterpretation when fluorescence lifetime is compared across prolonged periods (for example, days or weeks) or between samples with different sensor expression levels (for example, between individual animals). Thus, apparent changes in fluorescence lifetime observed over days, across cell types, or subcellular compartments may reflect variations in sensor expression levels rather than true differences in biological signals

(*Figure 6*). Therefore, considering biologically realistic factors in FLiSimBA is essential, as it qualitatively impacts the conclusions.

Importantly, our results not only highlight this issue but also provide a solution by defining the range where expression levels do not significantly affect lifetime (*Figures 5 and 6*). Practically, for a sensor with medium brightness delivered via in utero electroporation, adeno-associated virus, or as a knock-in gene, the brightness may not always fall within the expression level-independent regime. Therefore, these analyses are critical for identifying the conditions in which FLIM and FLiP can be reliably used to compare biosensor measurements over chronic time periods, across animals, and to quantify absolute levels of biological signals.

### Insights into future FLIM innovation

Our quantitative platform demonstrates how we can improve FLIM through future innovation. First, whereas our simulations showed little advantage of hybrid detectors over GaAsP PMTs in terms of the SNR (*Figure 4*), different types of hybrid detectors offer narrower IRFs at the cost of lower quantum efficiency (QE). FLiSimBA enables the evaluation of the trade-off between IRF width and QE. Additionally, hybrid detectors are more advantageous for sensors with shorter fluorescence lifetimes (*Trinh and Esposito, 2021*) and FLiSimBA allows exploration of the range in which they provide clear benefits in biological settings.

Second, we quantified how the development of red or far-red sensors outside the autofluorescence spectrum, along with minimizing detector dark currents, can significantly reduce or eliminate the expression level dependence of lifetime measurements (*Figure 7*). This opens exciting directions for sensor makers and hardware developers. The impact of these innovations on fluorescence lifetime consistency depends on the metrics used to measure lifetime (*Figure 7*). Fitted parameters, such as fitted $P_1$, are preferred over empirical lifetime when the lifetime histogram can be described by multi-exponential decay equations and sufficient photons are collected for fitting, because they are less sensitive to background interference (*Figure 7*). In contrast, empirical lifetime is used when sensor lifetime decay is too complex, or when photon counts are too low for fitting. FLiSimBA can be used to test multiple fitting methods and lifetime metrics (*Bajzer et al., 1991*; *Chen et al., 2022*; *Jo et al., 2005*; *Mannam et al., 2020*; *Periasamy, 1988*; *Rowley et al., 2011*; *Shen et al., 2024*; *Smith et al., 2019*; *Steinbach, 2012*; *Stringari et al., 2011*; *Wang et al., 2022*; *Wu et al., 2016*; *Xiao et al., 2023*; *Zickus et al., 2020*), and this will be an important future direction for identifying the best analysis method for specific experimental conditions.

Third, we demonstrate the feasibility and specify sensor parameters that enable the use of fluorescence lifetime and color spectrum as two orthogonal axes to distinguish many biological signals through intensity-based real-time imaging of biosensors (*Figure 8*). This approach promises to open doors to enhancing the ability of multiplexed imaging by orders of magnitude.

Thus, FLiSimBA and our quantitative framework are instrumental for enabling optimal experimental design, rigorous data interpretation, and pushing the limit of FLIM imaging through innovations.

## Materials and methods

**Key resources table**

| Reagent type (species) or resource | Designation | Source or reference | Identifiers | Additional information |
|---|---|---|---|---|
| Software, algorithm | Matlab | Matlab | RRID:SCR_001622 | |
| Software, algorithm | Python | Python.org | RRID:SCR_008394 | |
| Software, algorithm | Fluorescence Lifetime Simulation for Biological Applications (FLiSimBA) | Yao Chen's Laboratory (*Ma et al., 2025*) | | Authors: Pingchuan Ma, Peter Chen, Yao Chen, 2025, version 1, https://github.com/YaoChenLabWashU/Publication/tree/main/Simulation_manuscript |

### Animals

All procedures for rodent husbandry and surgery were performed following protocols approved by the Washington University Institutional Animal Care and Use Committee and in accordance with

National Institutes of Health guidelines. CD-1 mice (Envigo #030) were used. The experiments were performed according to the Animal Research: Reporting of In Vivo Experiments (ARRIVE) guidelines (*Percie du Sert et al., 2020*).

## DNA plasmid

For experimentally determined FLIM-AKAR data, AAV-FLIM-AKAR (*Chen et al., 2014*) (Addgene #63058) was used to express the FLIM-AKAR sensor in the primary somatosensory cortex by in utero electroporation (*Chen et al., 2017*).

## Acute brain slice preparation

Mice were anesthetized with isoflurane at 15–19 days of age and then decapitated. Their brains were rapidly dissected and put in sucrose-based cutting solution (concentrations in mM: 75 sucrose, 2.5 KCl, 1.25 $NaH_2PO_4$, 25 $NaHCO_3$, 87 NaCl, 25 glucose, 1 $MgCl_2$). 300 μm-thick coronal sections containing the primary somatosensory cortex were obtained with a vibratome (Leica Instruments, VT1200S) in cold sucrose-based cutting solution. After sectioning, the slices were transferred to artificial cerebral spinal fluid (ACSF) (concentrations in mM: 127 NaCl, 2.5 KCl, 1.25 $NaH_2PO_4$, 25 $NaHCO_3$, 2 $CaCl_2$, 1 $MgCl_2$, and 25 glucose) and incubated at 34°C for 10 min for recovery. The slices were kept at room temperature in ACSF with 5% $CO_2$ and 95% $O_2$. The slices were then transferred to a microscope chamber and ACSF was perfused at a flow rate of 2–4 mL/min for imaging.

## Two-photon fluorescence lifetime imaging microscopy (2pFLIM)

2pFLIM was performed as described previously (*Chen et al., 2014*; *Chen et al., 2017*; *Ma et al., 2024*; *Tilden et al., 2024*). A custom-built microscope with a mode-locked laser source (Spectra-Physics, Insight X3 operating at 80 MHz) was used. Photons were collected with fast photomultiplier tubes (PMTs, Hamamatsu, H10770PB-40). A 60 X (Olympus, NA 1.1) objective was used. Image acquisition was performed with the custom-written software ScanImage (*Chen et al., 2014*; *Chen et al., 2017*; *Ma et al., 2024*; *Pologruto et al., 2003*; *Tilden et al., 2024*) in MATLAB 2012b. The excitation wavelength was 920 nm. Emission light was collected through a dichroic mirror (FF580-FDi01−25X36, Semrock) and a band-pass filter (FF03-525/50-25, Semrock). Images covering 15 μm × 15 μm fields of view were collected at 128x128 pixels via a frame scan at 4 Hz. The FLIM board SPC-150 (Becker and Hickl GmbH) was used, and time-correlated single photon counting was performed with 256 time channels. Photons from 20 frames were pooled for fluorescence lifetime calculations. Only healthy cells (judged by gradient contrast images) at 30–50 μm below the slice surface were selected. Each individual cell was analyzed as a region of interest (ROI). Photons from a given ROI were pooled for further analysis.

## Experimental data collection, parameter determination, and simulation

The simulation packages were provided in MATLAB and Python. The simulation was performed in MATLAB 2022a or Python with the following steps (*Figure 1A*). The final simulated histograms consisted of the IRF-convolved sensor fluorescence, autofluorescence, afterpulse, and background fluorescence. The simulations were performed with 256 time channels for each laser cycle from an 80 MHz laser (12.5 ns interpulse interval). For each $P_1$ and sensor photon number condition, the simulation was repeated 500 times.

### Generation of photon populations for sensor fluorescence

For FLIM-AKAR sensor fluorescence, $\tau_1$ and $\tau_2$ were previously determined to be 2.14 ns and 0.69 ns, respectively (*Chen et al., 2014*; *Chen et al., 2017*). To determine the appropriate photon counts and $P_1$ range for simulation, sensor fluorescence histogram was fitted with *Equation 3*.

For *Figures 1–7*, to simulate the fluorescence lifetime with double exponential decay, we generated a population of photons with *Equation 1* with $F_0$ equal to 1,000,000. We generated photon populations and corresponding fluorescence lifetime histograms with $P_1$ ranging from 0.4 to 0.6 with an increment of 0.01.

### Sensor fluorescence sampling and IRF convolution

For sensor fluorescence sampling, a specific number of photons were randomly drawn with replacement from the corresponding population generated with double exponential decay. IRF convolution

of the fluorescence lifetime histogram was performed: the lifetime of each photon in the sample was redistributed along the time channels based on the probability of the IRF distribution. Following convolution, the histogram was wrapped around such that any photons whose lifetimes were beyond 12.5 ns (inter-pulse interval of laser) were redistributed to the next cycle.

In all the figures except for *Figure 4*, the IRF was empirically measured by second harmonic generation of the mouse tail with excitation at 1050 nm. The lifetime histogram was normalized to the total photon number and used as the IRF. For *Figure 4*, the IRFs of both systems were modeled as Gaussian distributions with different Gaussian widths. The mean of the Gaussian distribution (*μ*) was set as the peak channel of the experimentally collected IRF. The full width at half maximum (FWHM) of the Gaussian IRF of the GaAsP PMT was set to match the FWHM of the experimentally collected IRF (340 ps). The FWHM of the hybrid detector Gaussian IRF was set as 120 ps based on the HPM-100–40 model (Becker & Hickl) (*Becker et al., 2011*). The standard deviation (STD, *σ*) of the Gaussian distribution of the IRFs was determined based on the relationship with the FWHM:

$$FWHM = 2\sqrt{2ln2} * \sigma \tag{4}$$

The Gaussian distribution was defined as follows:

$$G\left(t\right) = \frac{1}{\sigma\sqrt{2\pi}} * e^{\frac{-1}{2}\left(\frac{t-\mu}{\sigma}\right)^2} \tag{5}$$

The Gaussian IRF was generated by normalization of *G(t)* against the total photon count.

For *Figure 4G and H*, to isolate the noise from the sampling of photons, sensor fluorescence sampling was performed 500 times, followed by analytical IRF convolution of each sensor fluorescence decay histogram. To isolate the noise from sampling from the IRF distribution, only one sensor photon decay histogram was generated, and each photon was reassigned to a single lifetime value based on the probability density distribution of the IRF, and this procedure was repeated 500 times.

## Autofluorescence

All biological tissues exhibit autofluorescence due to fluorescent cellular components and metabolites, such as nicotinamide-adenine dinucleotide (NAD), flavins, and aromatic amino acids (*Georgakoudi and Quinn, 2023*; *Ma et al., 2024*; *Malak et al., 2022*).

To determine the autofluorescence of brain tissue, we imaged acute brain slices from mice aged 15–19 days postnatal that did not express sensors. Neurons from the primary somatosensory cortex were imaged under the following conditions: 920 nm excitation light, 2.5 mW power, 30–50 μm below the slice surface, 20 frames pooled over 5 s, and a field of view of 15 μm x 15 μm. Fluorescence decay histograms from 19 images of two brain slices from a single mouse were averaged. These histograms included contributions from both autofluorescence and background signals (due to dark current and ambient light leakage). The average autofluorescence histogram was fitted with a double exponential function with background (*Equation 3*) to determine the parameters $\tau_1$, $\tau_2$, $P_1$, and $P_2$ of autofluorescence as well as background signals. The number of photons from autofluorescence ($F_{auto}$) was calculated as the remaining photons after subtracting the background signals ($F_{background}$) from the average histogram. In this study, $F_{auto}$ was determined to be 4560 photons and $F_{background}$ was 3484 photons.

For simulations, we introduced up to 10% fluctuation in the number of photons by randomly drawing an integer within the range of $F_{auto}*(1 \pm 5\%)$.

For *Figure 4*, the autofluorescence lifetime was simulated as double exponential decay (*Equation 1*). For the remaining figures, the autofluorescence lifetime was sampled from the empirical autofluorescence lifetime distribution, where the background was subtracted from the average autofluorescence lifetime histogram.

## Afterpulse and background fluorescence

The afterpulse ratio of the PMT was derived from the IRF histogram and the background fluorescence measurement described above. Then the photons per channel were averaged at the end of the IRF histograms, where the distribution was even across time channels. The number was subtracted by the background fluorescence and the result was used as the afterpulse. Subsequently, the ratio between

the number of photons contributing to the afterpulse and the total number of photons from the IRF histogram was calculated as the afterpulse ratio. For GaAsP PMTs, an afterpulse ratio of 0.32% was used in all figures except for *Figure 2* and *Figure 2—figure supplement 1*, where a ratio of 0.40% was used to match the specific experimental data.

Afterpulse and background fluorescence were simulated by sampling with replacement from an even distribution across time channels. The number of photons contributing to background fluorescence ($F_{background}$) was determined from autofluorescence fitting. Up to 10% fluctuation was introduced to the number of photons with the random draw of an integer within the range of $F_{background}*(1 \pm 5\%)$. For the low background conditions in *Figure 7*, the number of photons contributing to the background was half of the background photon count for the remaining figures. The number of photons from the afterpulse was determined by the afterpulse ratio (0.32% or 0.40% for GaAsP PMTs and 0 for hybrid detectors) multiplied by the number of photons from the sensor fluorescence.

Please note that we used the specific parameters described above for this study, including using the double-exponential decay model and parameters for FLIM-AKAR, the measured autofluorescence amount and distribution from acute mouse brain slices at a specific age, the empirically measured IRF, afterpulse ratio, and background fluorescence for our 2pFLIM microscope. These input parameters (*Figure 1*) should be altered to adapt to different biological applications. Although they do not change the conclusions of this study (e.g. there is an expression level-dependent and -independent regime for fluorescence lifetime), the specific input parameters would alter the quantitative thresholds (e.g. the precise threshold above which fluorescence lifetime is not significantly altered by expression levels).

## FLIM analysis

Two metrics were used for subsequent data visualization and analysis. First, after the fluorescence lifetime histograms were fitted with *Equation 3*, fitted $P_1$, corresponding to the proportion of slower decay (2.14 ns), was used for data visualization and analysis. Second, the empirical lifetimes of all the photons were calculated based on *Equation 2*.

For all figures except for *Figure 2* and *Figure 2—figure supplement 1*, the time range from 0.489 ns to 11.5 ns was used for both $P_1$ fitting and empirical lifetime calculations, with the first time channel within this range used as time 0.0488 ns (12.5 ns/256 time channels) in the calculation.

For comparison between experimental and simulated data, we determined the photon counts and $P_1$ of experimental conditions from fluorescence lifetime histograms from whole fields of view in brain slices. We subtracted empirically collected autofluorescence (which includes dark currents, SHG, and background) and afterpulse (with an afterpulse ratio of 0.4%) from the histogram to derive the sensor histogram. Sensor histograms from seven experimental acquisitions were averaged and fitted with *Equation 3*, using a lifetime range of 1.8–11.5 ns, with *SHG* and $F_{background}$ set to 0. The fitted $P_1$ (0.5854) and photon number (652,126) were used for simulations in *Figure 2B*, *Figure 2—figure supplement 1B and C*. To minimize system artifact interference at the histogram edges, fitted $P_1$ and empirical lifetime were calculated within the 1.8–11.5 ns range for *Figure 2* and *Figure 2—figure supplement 1B and C*.

## Quantification and statistical analysis

For each simulated condition, the mean and STD of the fitted $P_1$ or the empirical lifetime of the 500 simulation repeats were calculated.

For *Figures 3 and 4*, the minimum detectable difference (MDD) was calculated by:

$$MDD = z * SE(\widehat{diff}) \tag{6}$$

where SE refers to the standard error, and $\widehat{diff}$ is the estimated mean difference between two distributions. With a significance level of 0.05 and a power of 0.8, the z value is 2.806. which was used for the calculations in this study. Under a certain sensor photon number, the STD of the two metrics under different $P_1$ conditions (condition 1 and condition 2) were similar. Thus, given a certain sensor photon count, the STD under $P_1$ condition (0.5 in *Figure 2*) was used to calculate the MDD:

$$MDD = z * \frac{\sqrt{STD_{(condition\,1)}^2 + STD_{(condition\,2)}^2}}{\sqrt{n}} \backsim \frac{z * \sqrt{2} * STD}{\sqrt{n}} \tag{7}$$

where n is the number of data pairs used to analyze whether there was a significant change in the fitted $P_1$ or empirical lifetime.

For *Figures 6 and 7*, to determine whether a certain amount of sensor expression-induced apparent change in the fitted $P_1$ or empirical lifetime was significant, t tests were used to compare whether there was any statistically significant difference between two distributions with the same simulated $P_1$ but different photon numbers. The critical value of the t statistic is z=1.96 for a significance level of 0.05. The equation $z * SE(\widehat{diff})$ was used to calculate the fitted $P_1$ or empirical lifetime difference that would be statistically significant. As the photon count increased, there was an intersection where the mean fitted $P_1$ or empirical lifetime difference between the two distributions became less than the difference that would reach statistical significance, and this intersection point (determined by linear interpolation of the curves plotted on a log scale for both axes) was used to determine the minimum number of sensor photons required to tolerate a specific amount of change in sensor fluorescence.

Detailed information on the quantification, sample size, and statistical tests used are summarized in the Figure Legends, Figures, and Results. T tests were performed to test whether two distributions had equal means. For analysis of variance, one-way or two-way ANOVA was performed followed by multiple comparison tests.

## Simulation and analysis of multiplexed imaging with fluorescence intensity and lifetime data

For *Figure 8* and *Figure 8—figure supplement 1*, the simulations were performed with a laser cycle of 50 ns to capture the fluorescence decay curve of Sensor 2 with a longer lifetime constant. The fluorescence lifetime histograms of sensor 1 and sensor 2 were simulated as single exponential decays:

$$F(t) = F_0 e^{\frac{-t}{\tau}} \otimes IRF \tag{8}$$

where the decay constants $\tau$ for sensor 1 and sensor 2 are different.

To simulate the fluorescence lifetime with single exponential decay, we generated a population of photons with *Equation 8* with $F_0$ equal to 1,000,000. For sensor 1, the population was generated with $\tau$=2.0 ns. For sensor 2, the population was generated with $\tau$=2.5, 3.0, 3.5, 4.0, or 6.0 ns.

Fluorescence lifetime histograms of sensor 1 and sensor 2 were generated by sampling from the corresponding population, followed by IRF convolution. The histograms of sensor 1 and sensor 2 were added together as the combined histogram. Autofluorescence and background signals were simulated with the same probability density function as above but with 3.2-fold of photon numbers to accommodate for the longer acquisition time needed with a longer laser cycle to reach the same number of photons. Afterpulse was added proportional to the sensor photons as described above.

The combined histograms were fitted by a double exponential decay (*Equation 3*), with fixed $\tau_1$ and $\tau_2$ that were used for the simulation that generated the histograms. The fitting was performed with the Gauss-Newton nonlinear least-square fitting algorithm. $F_0$, $P_1$, and $P_2$ generated from the fitting were used to calculate the number of photons that contributed to sensor 1 and sensor 2, respectively.

The statistical power to detect the difference between two distributions of sensor intensity was calculated as follows:

$$\beta = 1 - normcdf\left(1.96 - \frac{\Delta I}{SE(\widehat{diff})}\right) \tag{9}$$

where *normcdf* is the cumulative distribution function of a normal distribution, $\Delta I$ is the intensity (photon number) change to be detected, SE refers to the standard error, and $\widehat{diff}$ is the estimator of the mean difference between two distributions.

The relationship between the statistical power $\beta$ and the intensity change ratio ($\Delta F/F_0$) was fitted to a 5-parameter logistic curve:

$$\beta_{(\Delta F/F_0)} = D + \frac{A - D}{\left(1 + \left(\frac{\Delta F/F_0}{C}\right)^B\right)^E} \tag{10}$$

where A, B, C, D, and E are constants in the equation. The intensity change ratio corresponding to a statistical power of 80% is determined by solving the equation with $\beta_{(\Delta F/F_0)} = 0.8$ and defined as the minimum detectable $\Delta F/F_0$.

## Acknowledgements

We thank Mikhail Berezin, Hanley Chiang, and the members of Yao Chen's lab, Tim Holy's lab, and Daniel Kerschensteiner's lab for helpful feedback on the study. We thank Kerry Grens, Chao Zhou, and Erwin Arias for critical comments on the manuscript. We thank Elizabeth Tilden for code review, and Aditi Maduska for sharing FLIM-AKAR imaging data. We thank the following sources for funding support for this work: U.S. National Institute of Neurological Disorders and Stroke (R01 NS119821, to YC), Brain & Behavior Research Foundation (NARSAD Young Investigator Grant 28323, to YC), Whitehall Foundation (2019-08-64, to YC), and McDonnell International Scholars Academy of Washington University in St. Louis (to PM).

## Additional information

### Funding

| Funder | Grant reference number | Author |
|---|---|---|
| National Institute of Neurological Disorders and Stroke | R01 NS119821 | Yao Chen |
| Brain and Behavior Research Foundation | NARSAD Young Investigator Grant 28323 | Yao Chen |
| Whitehall Foundation | 2019-08-64 | Yao Chen |
| McDonnell International Scholars Academy, Washington University in St. Louis | | Pingchuan Ma |

The funders had no role in study design, data collection and interpretation, or the decision to submit the work for publication.

### Author contributions

Pingchuan Ma, Conceptualization, Data curation, Software, Formal analysis, Validation, Investigation, Visualization, Methodology, Writing – original draft, Writing – review and editing; Peter Chen, Software, Validation, Methodology, Writing – review and editing; Scott Sternson, Conceptualization, Writing – review and editing; Yao Chen, Conceptualization, Resources, Software, Formal analysis, Supervision, Funding acquisition, Validation, Investigation, Visualization, Methodology, Writing – original draft, Project administration, Writing – review and editing

### Author ORCIDs

Pingchuan Ma (ID) https://orcid.org/0000-0001-9964-2852
Peter Chen (ID) https://orcid.org/0009-0008-8531-5509
Scott Sternson (ID) https://orcid.org/0000-0002-0835-444X
Yao Chen (ID) https://orcid.org/0000-0003-1509-6634

### Ethics

All procedures for rodent husbandry and surgery were performed following protocols approved by the Washington University Institutional Animal Care and Use Committee (#21-0390) and in accordance with the recommendations in the Guide for the Care and Use of Laboratory Animals of the National Institutes of Health. The experiments were performed according to the Animal Research: Reporting of In Vivo Experiments (ARRIVE) guidelines (Percie du Sert et al., 2020).

Reviewer #1 (Public review): https://doi.org/10.7554/eLife.101559.3.sa1

Reviewer #3 (Public review): https://doi.org/10.7554/eLife.101559.3.sa2
Author response https://doi.org/10.7554/eLife.101559.3.sa3

## Additional files

**Supplementary files**
MDAR checklist

**Data availability**

The current manuscript is a computational study. The MATLAB and Python functions for simulation and data analyses are deposited at GitHub (*Ma et al., 2025*).

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
