## [Editor Report · eLife Assessment]

This study presents an **important** computational framework, FLiSimBA (Fluorescence Lifetime Simulation for Biological Applications), for modeling experimental limitations in Fluorescence Lifetime Imaging Microscopy (FLIM). FLiSimBA is readily available in MATLAB and Python, enables users to simulate effects of noise and varying sensor expression levels, and provides practical guidance for both lifetime imaging experiments and biosensor development. The analyses are robust, and the evidence supporting the tool's utility in distinguishing between multiple lifetime signals is **compelling**, indicating strong potential for multiplexed dynamic imaging. However, users should also consider that the tool's effectiveness depends on the suitability of a two-component discrete exponential model.

---

## [Referee Report · Reviewer #1 (Public review)]

In this study, Ma et al. aimed to determine previously uncharacterized contributions of tissue autofluorescence, detector afterpulse, and background noise on fluorescence lifetime measurement interpretations. They introduce a computational framework they named "Fluorescence Lifetime Simulation for Biological Applications (FLiSimBA)" to model experimental limitations in Fluorescence Lifetime Imaging Microscopy (FLIM) and determine parameters for achieving multiplexed imaging of dynamic biosensors using lifetime and intensity. By quantitatively defining sensor photon effects on signal to noise in either fitting or averaging methods of determining lifetime, the authors contradict any claims of FLIM sensor expression insensitivity to fluorescence lifetime and highlight how these artifacts occur differently depending on analysis method. Finally, the authors quantify how statistically meaningful experiments using multiplexed imaging could be achieved.

A major strength of the study is the effort to present results in a clear and understandable way given that most researcher do not think about these factors on a day-to-day basis. Additionally, the model code is readily available in Matlab and Python, which should allow for open access to a larger community.

Overall, the authors' achieved their aims of demonstrating how common factors (autofluorescence, background, and sensor expression) will affect lifetime measurements and they present a clear strategy for understanding how sensor expression may confound results if not properly considered. This work should bring to awareness an issue that new users of lifetime biosensors may not be aware of and that experts, while aware, have not quantitatively determine the conditions where these issues arise. This work will also point to future directions for improving experiments using fluorescence lifetime biosensors and the development of new sensors with more favorable properties.

---

## [Referee Report · Reviewer #3 (Public review)]

Summary:

This study presents a useful computational tool, termed FLiSimBA. The MATLAB-based FLiSimBA simulations allow users to examine the effects of various noise factors (such as autofluorescence, afterpulse of the photomultiplier tube detector, and other background signals) and varying sensor expression levels. Under the conditions explored, the simulations unveiled how these factors affect the observed lifetime measurements, thereby providing useful guidelines for experimental designs. Further simulations with two distinct fluorophores uncovered conditions in which two different lifetime signals could be distinguished, indicating multiplexed dynamic imaging may be possible.

Strengths:

The simulations and their analyses were done systematically and rigorously. FliSimba can be useful for guiding and validating fluorescence lifetime imaging studies. The simulations could define useful parameters such as the minimum number of photons required to detect a specific lifetime, how sensor protein expression level may affect the lifetime data, the conditions under which the lifetime would be insensitive to the sensor expression levels, and whether certain multiplexing could be feasible.

Weaknesses:

The analyses have relied on a key premise that the fluorescence lifetime in the system can be described as a two-component discrete exponential decay. This means that the experimenter should ensure that this is the right model for their fluorophores a priori.

---

## [Author Response]

The following is the authors’ response to the original reviews.

**Reviewer #1 (Public review):**
In this study, Ma et al. aimed to determine previously uncharacterized contributions of tissue autofluorescence, detector afterpulse, and background noise on fluorescence lifetime measurement interpretations. They introduce a computational framework they named "Fluorescence Lifetime Simulation for Biological Applications (FLiSimBA)" to model experimental limitations in Fluorescence Lifetime Imaging Microscopy (FLIM) and determine parameters for achieving multiplexed imaging of dynamic biosensors using lifetime and intensity. By quantitatively defining sensor photon effects on signal-to-noise in either fitting or averaging methods of determining lifetime, the authors contradict any claims of FLIM sensor expression insensitivity to fluorescence lifetime and highlight how these artifacts occur differently depending on the analysis method. Finally, the authors quantify how statistically meaningful experiments using multiplexed imaging could be achieved.A major strength of the study is the effort to present results in a clear and understandable way given that most researchers do not think about these factors on a day-to-day basis. The model code is available and written in Matlab, which should make it readily accessible, although a version in other common languages such as Python might help with dissemination in the community. One potential weakness is that the model uses parameters that are determined in aspecific way by the authors, and it is not clear how vastly other biological tissue and microscope setups may differ from the values used by the authors.Overall, the authors achieved their aims of demonstrating how common factors(autofluorescence, background, and sensor expression) will affect lifetime measurements and they present a clear strategy for understanding how sensor expression may confound results if not properly considered. This work should bring to awareness an issue that new users of lifetime biosensors may not be aware of and that experts, while aware, have not quantitatively determined the conditions where these issues arise. This work will also point to future directions for improving experiments using fluorescence lifetime biosensors and the development of new sensors with more favorable properties.

We appreciate the comments and helpful suggestions. We now also include FLiSimBA simulation code in Python in addition to Matlab to make it more accessible to the community.

One advantage of FLiSimBA is that the simulation package is flexible and adaptable, allowing users to input parameters based on the specific sensors, hardware, and autofluorescence measurements for their biological and optical systems. We used parameters based on a FRETbased sensor, measured autofluorescence from mouse tissue, and measured dark count/after pulse of our specific GaAsP PMT in this manuscript as examples. In Discussion and Materials and methods, we now emphasize this advantage and further clarify how these parameters can be adapted to diverse tissues, imaging systems, and sensors based on individual experiments. We further explain that these input parameters will not affect the conclusions of our study, but the specific input parameters would alter the quantitative thresholds.

**Reviewer #2 (Public review):**
Summary:By using simulations of common signal artefacts introduced by acquisition hardware and the sample itself, the authors are able to demonstrate methods to estimate their influence on the estimated lifetime, and lifetime proportions, when using signal fitting for fluorescence lifetime imaging.Strengths:They consider a range of effects such as after-pulsing and background signal, and present a range of situations that are relevant to many experimental situations.Weaknesses:A weakness is that they do not present enough detail on the fitting method that they used to estimate lifetimes and proportions. The method used will influence the results significantly. They seem to only use the "empirical lifetime" which is not a state of the art algorithm. The method used to deconvolve two multiplexed exponential signals is not given.

We appreciate the comments and constructive feedback. Our revision based on the reviewer’s suggestions has made our manuscript clearer and more user friendly. We originally described the detail of the fitting methods in Materials and methods. Given the importance of these methodological details for evaluating the conclusions of this study, we have moved the description of the fitting method from Materials and methods to Results. In addition, we provide further clarification and more details of the rationale of using these different methods of lifetime estimates in Discussion to aid users in choosing the best metric for evaluating fluorescence lifetime data.

More specifically, we modified our writing to highlight the following.

(1) In Results, we describe that lifetime histograms were fitted to Equation 3 with the GaussNewton nonlinear least-square fitting algorithm and the fitted P

(2) In Results, we clarify that our simulation of multiplexed imaging was modeled with two sensors, each displaying a single exponential decay, but the two sensors have different decay constants. We also describe that Equation 3 with the Gauss-Newton nonlinear least-square fitting algorithm was used to deconvolve the two multiplexed exponential signals (Fig. 8)

**Reviewer #3 (Public review):**
Summary:This study presents a useful computational tool, termed FLiSimBA. The MATLAB-based FLiSimBA simulations allow users to examine the effects of various noise factors (such as autofluorescence, afterpulse of the photomultiplier tube detector, and other background signals) and varying sensor expression levels. Under the conditions explored, the simulations unveiled how these factors affect the observed lifetime measurements, thereby providing useful guidelines for experimental designs. Further simulations with two distinct fluorophores uncovered conditions in which two different lifetime signals could be distinguished, indicating multiplexed dynamic imaging may be possible.Strengths:The simulations and their analyses were done systematically and rigorously. FliSimba can be useful for guiding and validating fluorescence lifetime imaging studies. The simulations could define useful parameters such as the minimum number of photons required to detect a specific lifetime, how sensor protein expression level may affect the lifetime data, the conditions under which the lifetime would be insensitive to the sensor expression levels, and whether certain multiplexing could be feasible.Weaknesses:The analyses have relied on a key premise that the fluorescence lifetime in the system can be described as two-component discrete exponential decay. This means that the experimenter should ensure that this is the right model for their fluorophores a priori and should keep in mind that the fluorescence lifetime of the fluorophores may not be perfectly described by a twocomponent discrete exponential (for which alternative algorithms have been implemented: e.g., Steinbach, P. J. Anal. Biochem. 427, 102-105, (2012)). In this regard, I also couldn't find how good the fits were for each simulation and experimental data to the given fitting equation (Equation 2, for example, for Figure 2C data).

We thank the reviewer for the constructive feedback. We agree that the FLiSimBA users should ensure that the right decay equations are used to describe the fluorescent sensors. In this study, we used a FRET-based PKA sensor FLIM-AKAR to provide proof-of-principle demonstration of the capability of FLiSimBA. The donor fluorophore of FLIM-AKAR, truncated monomeric enhanced GFP, displays a single exponential decay. FLIM-AKAR, a FRET-based sensor, displays a double exponential decay. The time constants of the two exponential components were determined and reported previously (Chen, et al, Neuron (2017)). Thus, a double exponential decay equation with known τ_1_ and τ_2_ was used for both simulation and fitting. The goodness of fit is now provided in Supplementary Fig. 1 for both simulated and experimental data. In addition to referencing our prior study characterizing the double exponential decay model of FLIM-AKAR in Materials and methods, we have emphasized in Discussion the versality of FLiSimBA to adapt to different sensors, tissues, and analysis methods, and the importance of using the right mathematical models to describe the fluorescence decay of specific sensors.

Also, in Figure 2C, the 'sensor only' simulation without accounting for autofluorescence (as seen in Sensor + autoF) or afterpulse and background fluorescence (as seen in Final simulated data) seems to recapitulate the experimental data reasonably well. So, at least in this particular case where experimental data is limited by its broad spread with limited data points, being able to incorporate the additional noise factors into the simulation tool didn't seem to matter too much.

In the original Fig 2C, the sensor fluorescence was much higher than the contributions from autofluorescence, afterpulse, and background signals, resulting in minimal effects of these other factors, as the reviewer noted. This original figure was based on photon counts from single neurons expressing FLIM-AKAR. For the rest of the manuscript, photon counts were based on whole fields of view (FOV). Since the FOV includes cells that do not express fluorescent sensors, the influence of autofluorescence, dark currents, and background is much more pronounced, as shown in Fig. 2B.

Both approaches – using photon counts from the whole FOV or from individual neurons – have their justifications. Photon counts from the whole FOV simulate data from fluorescence lifetime photometry (FLiP), whereas photon counts from individual neurons simulate data from fluorescence lifetime imaging microscopy (FLIM). However, the choice of approach does not affect the conclusions of the manuscript, as a range of photon count values are simulated. To maintain consistency throughout the manuscript, we have revised the photon counts in this figure (now Supplementary Fig. 1C) to match those from the whole FOV.

Additionally, we have made some modifications in our analyses of Supplementary Fig. 1C and Fig. 2B, detailed in the “FLIM analysis” section of Materials and methods. For instance, to minimize system artifact interference at the histogram edges, we now use a narrower time range (1.8 to 11.5 ns) for fitting and empirical lifetime calculation.

**Reviewer #1 (Recommendations for the authors):**
(1) The authors report how autofluorescence was measured from "imaged brain slices from mice at postnatal 15 to 19 days of age without sensor expression." However, it remains unclear how many acute slices and animals were used (for example, were all 15um x 15um FOV from a single slice) and if mouse age affects autofluorescence quantification. Furthermore, would in vivo measurements have different autofluorescence conditions given that blood flow would be active? It would help if the authors more clearly explained how reliable their autofluorescence measurement is by clarifying how they obtained it, whether this would vary across brain areas, and whether in vitro vs in vivo conditions would affect autofluorescence.

We have added description in Materials and methods that for autofluorescence ‘Fluorescence decay histograms from 19 images of two brain slices from a single mouse were averaged.’ We have added in Discussion that users should carefully ‘measure autofluorescence that matches the age, brain region, and data collection conditions (e.g., ex vivo or in vivo) of their tissue…’, and emphasize that FLiSimBA offers customization of inputs, and it is important for users to adapt the inputs such as autofluorescence to their experimental conditions. We also clarify in Discussion that the change of input parameters such as autofluorescence across age and brain region would not affect the general insights from this study, but will affect quantitative values.

(2) Does sensor expression level issues arise more with in-utero electroporation compared to AAV-based delivery of biosensors? A brief comment on this in the discussion may help as most users in the field today may be using AAV strategies to deliver biosensors.

In our experience, in-utero electroporation results in higher sensor expression than AAV-based delivery, and so pose less concern for expression-level dependence. However, both delivery methods can result in expression level dependence, especially with a sensor that is not bright. We have added in Discussion ‘For a sensor with medium brightness delivered via in utero electroporation, adeno-associated virus, or as a knock-in gene, the brightness may not always fall within the expression level-independent regime.’

(3) Figure 1. Should the x-axis on the top figures be "Time (ns)" instead of "Lifetime (ns)"?Similarly in Figure 8A&B, wouldn't it make more sense to have the x-axis be Time not Lifetime?

The x-axis labels in Fig. 1 and Fig. 8A-8B have been changed to ‘Time (ns)’.

(4) Figure 2b: why is the empirical lifetime close to 3.5ns? Shouldn't it be somewhere between2.14 and 0.69?

In our empirical lifetime calculation, we did not set the peak channel to have a time of 0.0488 ns (i.e. the laser cycle 12.5 ns divided by 256 time channels). Rather, we set the first time channel within a defined calculation range (i.e. 1.8 ns in Supplementary Fig. 1B) to have a time of 0.0488 ns (i.e.). Thus, the empirical lifetime exceeds 2.14 ns and depends on the time range of the histogram used for calculation.

For Fig. 2B and Supplementary Fig. 1C, we have now adjusted the range to 1.8-11.5 ns to eliminate FLIM artifacts at the histogram edges in our experimental data, resulting in an empirical lifetime around 2.255 ns. In contrast, the range for calculating the empirical lifetime of simulated data in the rest of the study (e.g. Fig. 4D) is 0.489-11.5 ns, yielding a larger lifetime of ~3.35 ns.

We have clarified these details and our rationale in Materials and methods.

(5) Figure 2b: how come the afterpulse+background contributes more to the empirical lifetime than the autofluorescence (shorter lifetime). This was unclear in the results text why autofluorescence photons did not alter empirical lifetime as much as did the afterpulse/background.

With a histogram range from 1.8 ns to 11.5 ns used in Fig. 2B, the empirical lifetime for FLIM-AKAR sensor fluorescence, autofluorescence, and background/afterpulse are: 2-2.3 ns, around 1.69 ns, and around 4.90 ns. The larger difference of background/afterpulse from FLIM-AKAR sensor fluorescence leads to larger influence of afterpulse+background than autofluorescence. We have added an explanation of this in Results.

(6) One overall suggestion for an improvement that could help active users of lifetime biosensors understand the consequences would be to show either a real or simulated example of a "typical experiment" conducted using FLIM-AKAR and how an incorrect interpretation could be drawn as a consequence of these artifacts. For example, do these confounds affect experiments involving comparisons across animals more than within-subject experiments such as washing a drug onto the brain slice, and the baseline period is used to normalize the change in signal? I think this type of direct discussion will help biosensor users more deeply grasp how these factors play out in common experiments being conducted.

We have added the following in Discussion, ‘…While this issue is less problematic when the same sample is compared over short periods (e.g. minutes), It can lead to misinterpretation when fluorescence lifetime is compared across prolonged periods or between samples when comparison is made across chronic time periods or between samples with different sensor expression levels. For example, apparent changes in fluorescence lifetime observed over days, across cell types, or subcellular compartments may actually reflect variations in sensor expression levels rather than true differences in biological signals (Fig. 6), Therefore, considering biologically realistic factors in FLiSimBA is essential, as it qualitatively impacts the conclusions.’

**Reviewer #2 (Recommendations for the authors):**
The paper would be improved with more detail on the fitting methods, and the use of state-of-theart methods. Consult for example the introduction of this paper where many methods are listed: https://www.mdpi.com/1424-8220/22/19/7293

We have moved the description of the Gauss-Newton nonlinear least-square fitting algorithm from Materials and methods to Results to enhance clarity. We appreciate the reviewer’s suggestion to combine FLiSimBA with various analysis methods. However, the primary focus of our manuscript is to call for attention of how specific contributing factors in biological experiments influence FLIM data, and to provide a tool that rigorously considers these factors to simulate FLIM data, which can then be used for fitting. Therefore, we did not expand the scope of our manuscript. Instead, we have added in the Discussion that ‘‘FLiSimBA can be used to test multiple fitting methods and lifetime metrics as an exciting future direction for identifying the best analysis method for specific experimental conditions’, citing relevant references.

I would also improve the content of the GitHub repository as it is very hard to identify to source code used for simulation and fitting.

We have reorganized and relabeled our GitHub repository and now have three folders labeled as ‘Simulation_inMatlab’, ‘DataAnalysis_inMatlab’, and ‘SimulationAnalysis_inPython’. We also updated the clarification of the contents of each folder in the README file.

**Reviewer #3 (Recommendations for the authors):**
(1) P. 10 "For example, to detect a P1 change of 0.006 or a lifetime change of 5 ps with one sample measurement in each comparison group, approximately 300,000 photons are needed." If I am reading the graphs in Figures 3B and C, this sentence is talking about the red line. However, the intersection of 0.006 in the MDD of P1 in 3B and red is not 3E5 photons. And the intersection of 0.005 ns and red in 3C is not 3E5 photons either. Are you sure you are talking about n=1? Maybe the values are correct for the blue curve with n=5.

Thank you for catching our error. We have corrected the text to ‘with five sample measurements’.

(2) Figure 2 (B) legend: It would be helpful to specify what is being compared in the legend. For example, consider revising "* p < 0.05 vs sensor only; n.s. not significant vs sensor + autoF; # p < 0.05 vs sensor + autoF. Two-way ANOVA with Šídák's multiple comparisons test" to "* p <0.05 for sensor + auto F (cyan) vs sensor only; n.s. not significant for final simulated data (purple) vs sensor + autoF; # p < 0.05 for final simulated data (purple) vs sensor + autoF. Twoway ANOVA with Šídák's multiple comparisons test".

We’ve made the change and thanks for the suggestion to make it clearer.

(3) Figure 2 (c) Can you please show the same Two-way ANOVA test values for Experimental vs. Sensor only and for Experimental vs. Sensor + autoF? Currently, the value (n.s.) is marked only for Experimental vs. Final simulation. Given that the experimental data are sparse (compared to the simulations), it seems likely that there may be no significant difference among the 3 different simulations regarding how well they match the experimental data. Also, can you specify the P1 and P2 of the experimental data used to generate the simulated data on this panel? Also, what is the reason why P1=0.5 was used for panels A and B, instead of the value matching the experimental value?

As the reviewer suggested, we have included statistical tests in the figure (now Supplementary Fig. 1C). Please see our response to the Public Review of Reviewer 3’s comments as well as our changes in Materials and Methods on other changes and their rationale for this figure. We have now specified the P_1_ value of the experimental data used to generate the simulated data on this panel both in Figure Legends and Materials and Methods. Based on the suggestion, we have now used the same P_1_ value in Fig. 2B.